# Genetic code expansion enables visualization of *Salmonella* type three secretion system components and secreted effectors

Moirangthem Kiran Singh[1,2], Parisa Zangoui[1,2], Yuki Yamanaka[1†], Linda J Kenney[1,2]*

[1]Mechanobiology Institute, T-Lab, 5A Engineering Drive 1, National University of Singapore, Singapore, Singapore; [2]Biochemistry and Molecular Biology, University of Texas Medical Branch, Galveston, United States

**Abstract** Type three secretion systems enable bacterial pathogens to inject effectors into the cytosol of eukaryotic hosts to reprogram cellular functions. It is technically challenging to label effectors and the secretion machinery without disrupting their structure/function. Herein, we present a new approach for labeling and visualization of previously intractable targets. Using genetic code expansion, we site-specifically labeled SsaP, the substrate specificity switch, and SifA, a here-to-fore unlabeled secreted effector. SsaP was secreted at later infection times; SsaP labeling demonstrated the stochasticity of injectisome and effector expression. SifA was labeled after secretion into host cells via fluorescent unnatural amino acids or non-fluorescent labels and a subsequent click reaction. We demonstrate the superiority of imaging after genetic code expansion compared to small molecule tags. It provides an alternative for labeling proteins that do not tolerate N- or C-terminal tags or fluorophores and thus is widely applicable to other secreted effectors and small proteins.

*For correspondence:
likenney@utmb.edu

Present address: †Nippon Dental University School of Life Dentistry at Tokyo, Tokyo, Japan

Competing interests: The authors declare that no competing interests exist.

## Introduction

*Salmonellae* use an evolved complex protein secretion system known as a type three secretion system (T3SS) to deliver virulence factors (referred to as effectors) directly into host cells (*Marlovits et al., 2004*; *Kubori et al., 1998*). The T3SS is a 'needle-like', complex, supramolecular structure that spans the bacterial inner and outer membrane and is capable of injecting effectors into the host (*Marlovits et al., 2004*; *Kubori et al., 1998*; *Erhardt et al., 2010*). Inside the host cytoplasm, effectors manipulate host cell signaling pathways to drive pathogenesis (*LaRock et al., 2015*). *Salmonella* uses two distinct T3SS that are encoded within *Salmonella* pathogenicity islands 1 and 2 that function at different times during infection. The *Salmonella* pathogenicity island 1 (SPI-1)-encoded T3SS delivers effectors that alter host actin cytoskeleton dynamics, leading to extensive plasma membrane ruffling and bacterial uptake (*LaRock et al., 2015*; *Galán and Waksman, 2018*). Upon entry into host cells, *Salmonella* resides within an acidic membrane-bound compartment, the *Salmonella*-containing vacuole (SCV). The acid pH of the SCV acidifies the bacterial cytoplasm and activates *Salmonella* pathogenicity island 2 (SPI-2)-encoded T3SS (*Liew et al., 2019*; *Kenney, 2019*) to translocate a cocktail of ~30 effectors across the vacuolar membrane into the host cytosol (*Erhardt et al., 2010*; *LaRock et al., 2015*; *Galán and Waksman, 2018*; *Liew et al., 2019*; *Kenney, 2019*; *Jennings et al., 2017*). This complex mixture of effectors induces massive remodeling of endosomes, leading to the formation of highly dynamic, extensive tubular membrane structures known as *Salmonella*-induced filaments (SIFs) (*LaRock et al., 2015*; *Galán and Waksman, 2018*;

*Jennings et al., 2017*; *Garcia-del Portillo et al., 1993*; *Drecktrah et al., 2008*; *Steele-Mortimer, 2008*; *Knuff and Finlay, 2017*). SifA is one of the key effectors that is involved in SIF formation (*Stein et al., 1996*); a *sifA* deletion eliminates SIF formation and attenuates virulence in the mouse (*Beuzón et al., 2000*; *Brumell et al., 2001*; *Ruiz-Albert et al., 2002*). To understand how such effectors manipulate host cell biology, it is informative to examine the localization of SPI-2 effectors within host cells.

Labeling and localization of secreted T3SS effectors in the host has proven to be technically challenging (see *Singh and Kenney, 2021* for a review). Earlier studies localized effectors by immunofluorescence of individually over-expressed effectors (*Ohlson et al., 2008*; *McGourty et al., 2012*). Not only was the stoichiometry of effectors disrupted by this approach, but also the effectors were not present at native copy numbers. Further, antibodies often produce off-target staining, and the location of epitope tag integration can be problematic (*Schnell et al., 2012*). The most conventional method for tracking and localization of proteins is to express the target protein fused to a fluorescent protein such as GFP or one of its variants. However, direct labeling of effectors with bulky fluorescent proteins interferes with secretion and jams the T3SS (*Akeda and Galán, 2005*). The utility of the 4Cys-FLaSH tagging system is limited by low labeling specificity, a poor signal-to-noise ratio, and cellular toxicity (*Enninga et al., 2005*; *Martin et al., 2005*; *Adams et al., 2002*). Tandem-repeat fluorescent tags such as split-GFP are also of limited value, because they do not overcome the dim signal from effectors expressed at low levels (*Young et al., 2017*; *Park et al., 2017*). All these methods require an additional protein tag fused to either the N- or C-terminus of the effector, and these tags, whether small or large, can perturb effector structure-function by disrupting trafficking or post-translational modification of fused effector proteins. Although the SPI-2 effector SseJ tolerated an HA-tag at its C-terminus (*Chakraborty et al., 2015*; *Gao et al., 2018*), we were unable to label SifA at either the N- or C-terminus or in intracellular loops and still preserve its function (*Gao et al., 2018*; *Brumell et al., 2002*). New strategies are therefore required to selectively label and visualize T3SS effectors in host cells during infection.

The T3SS systems share striking homology with the flagellar system and are believed to have evolved from a common ancestor (*Pallen and Gophna, 2007*). In the flagellar system, FliK functions as a 'ruler' protein to measure the hook length before switching to secrete the filament (*Shibata et al., 2007*). In pathogens such as *Yersinia*, *Shigella*, and *Salmonella*, FliK homologues control injectisome needle length (*Journet et al., 2003*; *Wee and Hughes, 2015*); its deletion produces long needles (*Muramoto et al., 1998*; *Kubori et al., 2000*; *Magdalena et al., 2002*; *Bergeron et al., 2016*). In the SPI-2 T3SS, SsaP is proposed to function in a similar manner (*Yu et al., 2018*), although its deletion does not produce long needles (Y. Yamanaka and L.J. Kenney, data not shown). SsaP is required for the secretion-specificity switch from substrates for injectisome assembly to the secretion of SPI-2 effectors, but its small size (124 amino acids) and low abundance makes labeling and visualization difficult. A C-terminal fusion with photoactivatable mCherry (PAmCherry) resulted in a fusion protein that was cleaved, preventing imaging of SsaP. How SsaP functions to control SPI-2 effector secretion is still unknown (*Yu et al., 2018*).

Genetic incorporation of non-canonical amino acids (ncAA) bearing bio-orthogonal chemical labeling handles such as azide, ring-strained alkynes, or alkenes into bacteria via genetic code expansion (GCE) provides a solution to these problems. A ncAA is first encoded by a nonsense codon (typically TAG) and is co-translationally incorporated into a protein of interest site-specifically, using a ncAA-specific orthogonal tRNA/aminoacyl-tRNA synthetase (tRNA/aaRS) pair (*Lang and Chin, 2014a*; *Davis and Chin, 2012*; *Chin et al., 2002*). Subsequently, the bacterial protein bearing a bio-orthogonal handle (such as azide, ring-strained alkenes) reacts with dibenzocyclooctyne (DBCO)/tetrazine (Tz)-containing fluorophores through catalyst free strain-promoted azide-alkyne cycloaddition (SPAAC) or strain-promoted inverse-electron–demand Diels–Alder cycloaddition (SPIEDAC) reactions (*Lang and Chin, 2014a*; *Agard et al., 2004*; *Plass et al., 2012*) as illustrated in *Figure 1* . Due to its high selectivity, fast reaction rate compared to SPAAC, and fluorogenic tetrazine fluorophore, the SPIEDAC click reaction has emerged as an invaluable selective bioconjugation tool in cells (*Plass et al., 2012*; *Lang and Chin, 2014b*). Since such a labeling approach is not limited to the N- or C-terminus of the effector of interest, and organic fluorophores are much smaller compared to fluorescent proteins (~0.5 vs. ~27 kDa), this approach reduces the potential negative impact of the fluorescent tags on the structure and function of labeled secreted effectors. The organic

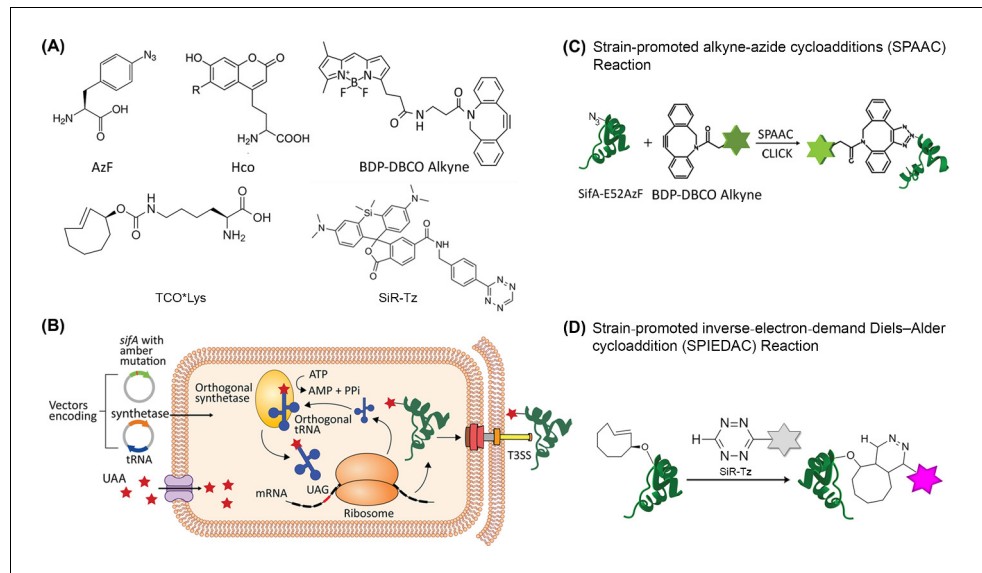

**Figure 1.** Scheme for site-specific fluorescent labeling of SPI-2 effectors. (**A**) Chemical structure of 4-azidophenylalanine (AzF), 7-(Hydroxy-coumarin-4-yl) ethylglycine (Hco), trans-Cyclooct-2-en – L - Lysine (TCO*Lys), BDP-DBCO, and SiR-tetrazine. (**B**) Incorporation of a ncAA in a SPI-2 effector is schematically represented. A plasmid carrying the gene for an effector of interest (green) and an orthogonal suppressor tRNA (dark blue)/aminoacyl synthetase (orange) pair are introduced into the bacterial cell by transformation. The amber stop codon (TAG, red) replaces a native codon at a permissive site within the sequence of the effector gene. The ncAA (red stars) is supplemented in the growth medium. Inside the cell, the orthogonal synthetase specifically charges the orthogonal suppressor tRNA with the desired ncAA through a catalytic reaction driven by ATP. The ncAA-acylated tRNA which contains a CUA anticodon, then enters the ribosomal machinery and incorporates the attached ncAA into the effector in response to the complementary amber codon on the effector mRNA (black). Once released from the ribosome, the tRNA can be further reused for ncAA amino-acylation by the cognate synthetase. The full-length polypeptide chain of the effector site-specifically carries the ncAA and undergoes folding and assembly into a functioning effector protein. The newly formed effector is translocated into the host cell through the T3SS. The secreted SPI-2 effectors incorporated with ncAA can be labeled by an externally added fluorophore. (**C**) Reaction scheme of effector labeling with a genetically encoded azide-containing protein. An azide-containing amino acid (AzF) is genetically incorporated into a protein (SifA) and the azido group reacts with a conjugation reagent containing dibenzocyclooctyne (BDP-DBCO) through SPAAC. (**D**) Reaction scheme for the copper-free click reaction with a fluorogenic tetrazine dye. An ncAA with a strained alkene group (e.g. TCO-Lys, as shown here) incorporated into an effector protein (SsaP) reacts with tetrazine-coupled dyes (SiR-Tz) through SPIEDAC click reaction. Dyes coupled to tetrazine are only fluorescent (magenta) after successful labeling.

fluorophores can also offer superior photophysical properties and more spectral variants compared to fluorescent proteins.

Although GCE has been used for specific labeling of cellular proteins, protein-protein interactions and post-translational studies in *E. coli*, yeast and mammalian cells (*Lang and Chin, 2014a*; *Davis and Chin, 2012*; *Chin et al., 2002*; *Lin et al., 2011*; *Peng and Hang, 2016*; *Ko et al., 2016*; *Zhang et al., 2011*; *Lammers et al., 2010*; *Hancock et al., 2010*), no study has used GCE for direct visualization of secreted bacterial effector proteins inside host cells. Herein, we report a GCE-based approach for fluorescent labeling and visualization of the secreted effector SifA of *Salmonella enterica* serovar Typhimurium, as well as the substrate specificity switch protein SsaP (*Figure 1C–D*). The efficiency and specificity of this labeling strategy is notably affected by the cellular properties of the organic fluorophores. One solution to this problem is to visualize secreted SifA directly by site-specifically introducing a relatively compact fluorescent ncAA (7-(Hydroxy-coumarin-4-yl) ethylglycine, Hco), avoiding the subsequent click reaction. The versatility of the GCE-based labeling approach allowed us to use it in fixed-cells, live-cells, as well as super-resolution imaging of bacterial secreted effectors inside host cells. Using this approach, we observed that SifA was co-localized with LAMP1, the effector SseJ, and the host motor protein kinesin. However, in contrast to SseJ, SifA did not continuously decorate the SIFs, indicating that the stoichiometry of SifA to SseJ was not 1:1. SsaP was

labeled and visualized for the first time in bacterial cells. It was localized predominantly at the cell poles, as was the SPI-2 T3SS apparatus (*Chakraborty et al., 2015*). SsaP was secreted into host cells at later times during infection. Finally, we provide evidence of the superiority of labeling by GCE compared to small molecule HA tags. The approach and application described herein is not limited to SifA, SseJ, or SsaP, but is readily applicable to other T3SS effector proteins, as well as to proteins that are not secreted, yet difficult to label.

## Results

### Engineering and selection of label positions

Theoretically, an ncAA can be incorporated into any site of a protein of interest. However, in practice, finding a suitable site can be laborious and empirical. Although the incorporation efficiency of an ncAA by an orthogonal tRNA/aminoacyl-tRNA synthetase (tRNA/aaRS) system is influenced by multiple factors, codon context appears to be the most important (*Xu et al., 2016*). The preferred context of orthogonal tRNA is AAT-TAG-ACT, affording the most efficient ncAA incorporation (*Xu et al., 2016*). *Salmonella* Typhimurium LT2 contains approximately 452 endogenous TAG stop codons out of 4696 genes, i.e., 9.6% of its genome (*Nakamura et al., 2000*). To the best of our knowledge, only a few SPI-1 genes (SopA) and SPI-2 genes (PipB, PipB2 and SspH2) employ TAG codons. Given that nearly 10% of the genome contains a TAG codon, one would anticipate a certain level of ncAA incorporation at endogenous TAGs, which would lead to spurious dye labeling of other proteins, in addition to labeling of the protein of interest. However, in-gel fluorescence of off-target incorporation into the proteomes of an *E. coli* derivative of BW25993 (where all the TAG codons were mutated to TAA stop codons) was compared with its wild-type parent. Similar levels of 'off-target' incorporation in the presence of the orthogonal tRNA and ncAA were observed (*Kipper et al., 2017*; *Uttamapinant et al., 2015*). We further circumvent this problem by examining SifA and SsaP under SPI-2-inducing conditions, during which SPI-1 effectors are present at very low (if any) concentrations. Since the tRNA preferred sequence is actually AAT-TAG-ACT (*Xu et al., 2016*), this sequence is absent in the TAG-containing SPI-1 and SPI-2 genes. Lastly, TAG codons tend to be present in genes that are not abundantly expressed (*Kipper et al., 2017*). For all the above reasons, amber suppression and subsequent labeling of off target proteins is quite rare.

### Site-specific incorporation of labels into SifA and SsaP

We selected the secreted effector SifA and SsaP for labeling by GCE, because of the past difficulties in labeling (*Gao et al., 2018*; *Brumell et al., 2002*). In the case of SsaP, it is a small protein (124 amino acids) and labeling with photoactivatable mCherry (PAmCherry) resulted in a cleaved fusion protein (*Figure 2—figure supplement 1*). The ncAA contains an azide group/ring-strained alkenes (TCO), thus SifA and SsaP can react with the DBCO/Tz-containing dyes by genetically incorporating AzF/TCO*Lys into its permissive sites. We first identified potential amino acid residues that would be optimal candidates for ncAA incorporation into SifA using amino acid conservation analysis with the ConSurf web server (*Ashkenazy et al., 2010*). We chose position 52 in SifA for AzF incorporation based on its low conservation score and the surface accessibility of the selected position was verified from the protein structure (*Figure 2—figure supplement 2*). In the case of SsaP, we engineered the TAG at position 65 to ensure a higher likelihood of labeling (*Xu et al., 2016*), this position was also within a region that had tolerated the insertion of an HA tag (*Figure 2—figure supplement 3*). Expression of either the full-length SifA-E52AzF mutant or SsaP-Y65TCO was dependent on the presence of ncAAs and the corresponding optimal tRNA/aaRS, as evident from SDS-PAGE (*Figure 2*). SifA-E52AzF and SsaP-Y65TCO were labeled with BDP-DBCO and SiR-Tz using SPAAC/SPIE-DAC click chemistry, respectively. A bright fluorescent band was present that corresponded to full length AzF-SifA (*Figure 2A*). In the case of SsaP-Y65TCO, a weak, diffusive fluorescent band was observed (*Figure 2C*) that was confirmed by western blot analysis of secreted SsaP (*Figure 2—figure supplement 3*). Fluorescence microscopy analysis further verified the selective labeling of SifA/SsaP in *Salmonella* cells in vitro (*Figure 2B,D*).

To evaluate the specificity of ncAA incorporation in SifA, we compared images of SifA-GFP that lacked the TAG codon (*sifA*-WT-GFP), or contained the TAG (*sifA*-TAG-GFP) (*Figure 2—figure supplement 4*). In the *sifA*-WT-GFP, the GFP signal was apparent (top panel), but no labeling by

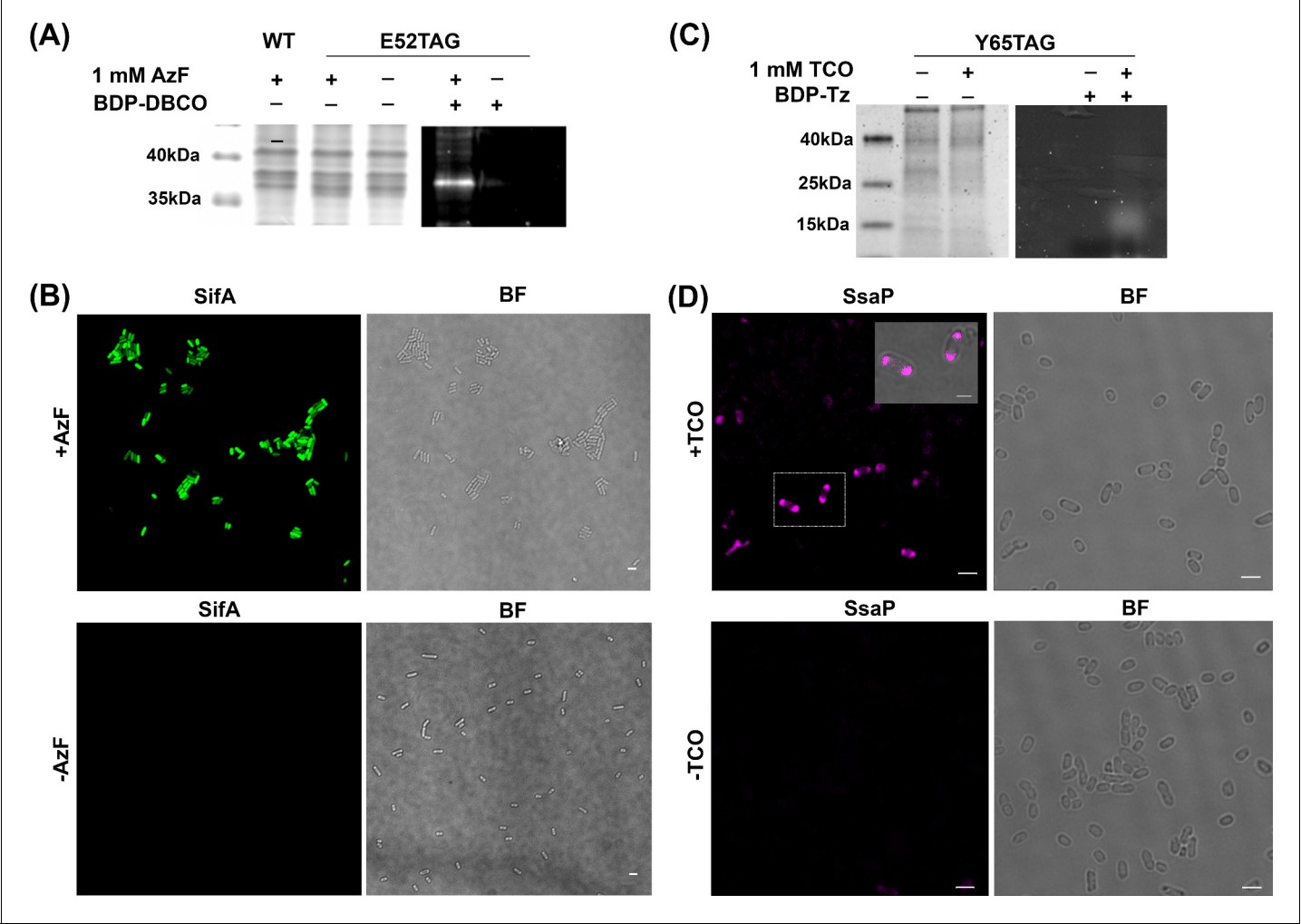

**Figure 2.** AzF is site-specifically incorporated into SifA-E52TAG in *Salmonella*. (A) Left: Coomassie stained SDS-PAGE; right: SDS-PAGE fluorescence imaging confirms selective labeling of SifA-E52AzF with BDP-DBCO in the cell lysate. (B) Expression of SifA-E52AzF in *Salmonella* analyzed by fluorescence microscopy in the presence (top) or absence (bottom) of 1mM AzF. *Salmonella* cells expressing SifA-E52AzF in the presence or absence of AzF were incubated with BDP-DBCO, and imaged for BDP fluorescence (green). SifA fluorescence (green) is only observed in the presence of AzF. (C) Site-specific incorporation of TCO*Lys into SsaP-Y65TCO in *Salmonella* left: Coomassie stained SDS-PAGE; right: SDS-PAGE fluorescence imaging confirms selective labeling of SsaPY65TCO with BDP-Tz in the secreted fraction collected after TCA precipitation. (D) Expression of SsaP-Y65TCO in *Salmonella* analyzed by fluorescence microscopy in the presence (top) or absence (bottom) of 1mM TCO*Lys. *Salmonella* cells expressing SsaP-Y65TCO in the presence or absence of TCO*Lys were incubated with SiR-Tz, fixed at 8 hr post acid induction and imaged for SiR fluorescence (magenta). Polarly localized SsaP fluorescence (magenta) is only observed in the presence of TCO*Lys. A higher magnification merged imaged is shown in the inset. See *Figure 2—figure supplement 5* for statistical analysis for the fraction of polarly localized SsaP. Images were acquired using confocal microscopy. The data are representative of at least three independent experiments. BF = bright field. Scale bar, 2 μm (B,D), 1 μm (inset).

The online version of this article includes the following source data and figure supplement(s) for figure 2:

**Figure supplement 1.** SsaP is required for SPI-2 secretion.

**Figure supplement 2.** Selection of ncAA incorporation sites.

**Figure supplement 3.** Western blot of SseB (translocon) and SsaP-HA secretion.

**Figure supplement 4.** AzF and subsequent Alexa555 sDIBO Alkyne incorporation is highly specific.

**Figure supplement 5.** Average fraction of cells that display polarly localized SsaP ($n_{total}$ = 112 cells).

**Figure supplement 5—source data 1.** Source data for *Figure 2—figure supplement 5*.

Alexa555 sDIBO was observed (bottom). In the case of *sifA*-TAG-GFP, in the absence of AzF, there was no labeled SifA (middle top panel), but addition of AzF restored amber suppression and the GFP signal was observed (top right). Labeling by Alexa555 sDIBO of SifA was identical to the SifA GFP image (bottom right). This experiment demonstrated the high specificity of ncAA incorporation

and the absence of off-site labeling. Thus, ncAAs were site-specifically incorporated at an amber codon in *sifA* and *ssaP* using GCE.

## Amber suppression does not affect *Salmonella* pathogenesis

To determine the consequences of the pEVOL orthogonal amber suppression system on *Salmonella* pathogenesis, we first evaluated the effect of pEVOL on cell growth of wild-type *Salmonella* and its isogenic *sifA* null strain. The presence of pEVOL had no significant effect on bacterial cell growth (*Figure 3—figure supplement 1A*). We next evaluated the effect of pEVOL on the ability of *Salmonella* to survive inside HeLa cells (*Figure 3—figure supplement 1B*). The survival of the pEVOL transformant was 74% compared to the wild-type which lacked the plasmid, indicating that intracellular survival of *Salmonella* was not significantly affected by the presence of pEVOL. Taken together, these results indicate that the orthogonal tRNA/aaRS amber suppression system pEVOL has no negative consequences on *Salmonella* pathogenesis.

## Engineered SifA rescues SIF formation

To determine whether the AzF-incorporated SifA could be translocated into host cells, we infected HeLa cells with the Δ*sifA* strain complemented with p*sifA*52TAG (see Materials and methods). At 16 hr post-infection, cells were fixed with paraformaldehyde (PFA) and immuno-stained with anti-LAMP1 antibody to reveal the SIFs. As shown in *Figure 3A*, SifA-E52AzF was clearly secreted, SIFs were formed, and the SIFs resembled those observed in wild-type infected cells (*Figure 3—figure supplement 2*). These observations confirmed that SIFs were rescued by the expression of *sifA*52-TAG using GCE. It also indicated that secreted engineered SifA-E52AzF was a fully functional virulence factor in *Salmonella*.

## Visualization of injected SifA into host cells by fluorescence imaging

Once we established that GCE-labeled SifA was functional (*Figure 3A*), the next step was to visualize secreted SifA-E52AzF labeled with BDP-DBCO via the SPAAC reaction in HeLa cells (*Agard et al., 2004*). After labeling, excess dye was washed out and cells were fixed with PFA and directly analyzed by spinning-disk Structured-illumination microscopy (SIM). SifA was secreted into the cytoplasm of HeLa cells and labeled with the fluorophore (*Figure 3B,D* and *Figure 4*). Intracellular fluorescent puncta were clearly observed in BDP-DBCO channels when SifA-E52AzF was expressed in the presence of AzF (*D'Costa et al., 2019*). In the absence of AzF, non-specific fluorescence signals were nearly absent (*Figure 3C*), further validating the specificity of our bio-orthogonal imaging of SifA in host cells.

To establish that the fluorescence signal we observed in HeLa cells was specific to the secreted effector SifA, and not from a SPI-2 effector that contains a TAG codon (e.g. PipB, PipB2), we infected HeLa cells with wild-type *Salmonella* (containing HA-tagged SseJ) in the presence of pEVOL and ncAA. HeLa cells were treated with click dye to visualize any ncAA-incorporated effectors that were secreted. After labeling, excess dye was washed out and cells were fixed with 4% PFA and immuno-stained with anti-HA (SseJ) and anti-LPS antibodies (*Salmonella*) after 16 hr post-infection. SseJ was secreted into the host cytoplasm, and intracellular and non-specific fluorescence signals were virtually absent (*Figure 3—figure supplement 3*). This result established that the fluorescence signal was specific to SifA and not coming from other SPI-2 secreted proteins. Thus, using GCE, we could visualize SifA within SIFs for the first time, and AzF labeling can detect and visualize injected virulence factors of bacterial pathogens.

## SifA co-localizes with LAMP1, the secreted effector SseJ and the host motor protein kinesin in SIFs

Once we established the optimal conditions for labeling SifA (*Figure 3*), we sought to interrogate the localization of SifA with respect to the SIFs using fluorescence imaging. Infected HeLa cells were labeled with the fluorophore BDP-DBCO in the presence of 1 mM AzF and prepared as before (*Figure 3B*) and SIFs were stained with fluorophore-conjugated anti-LAMP1 antibody for confocal fluorescence imaging (*Figure 3D*). It is evident from the merged image that SifA co-localizes with the lysosomal marker LAMP1 (*Figure 3D*), and firmly establishes the presence of SifA within the SIFs.

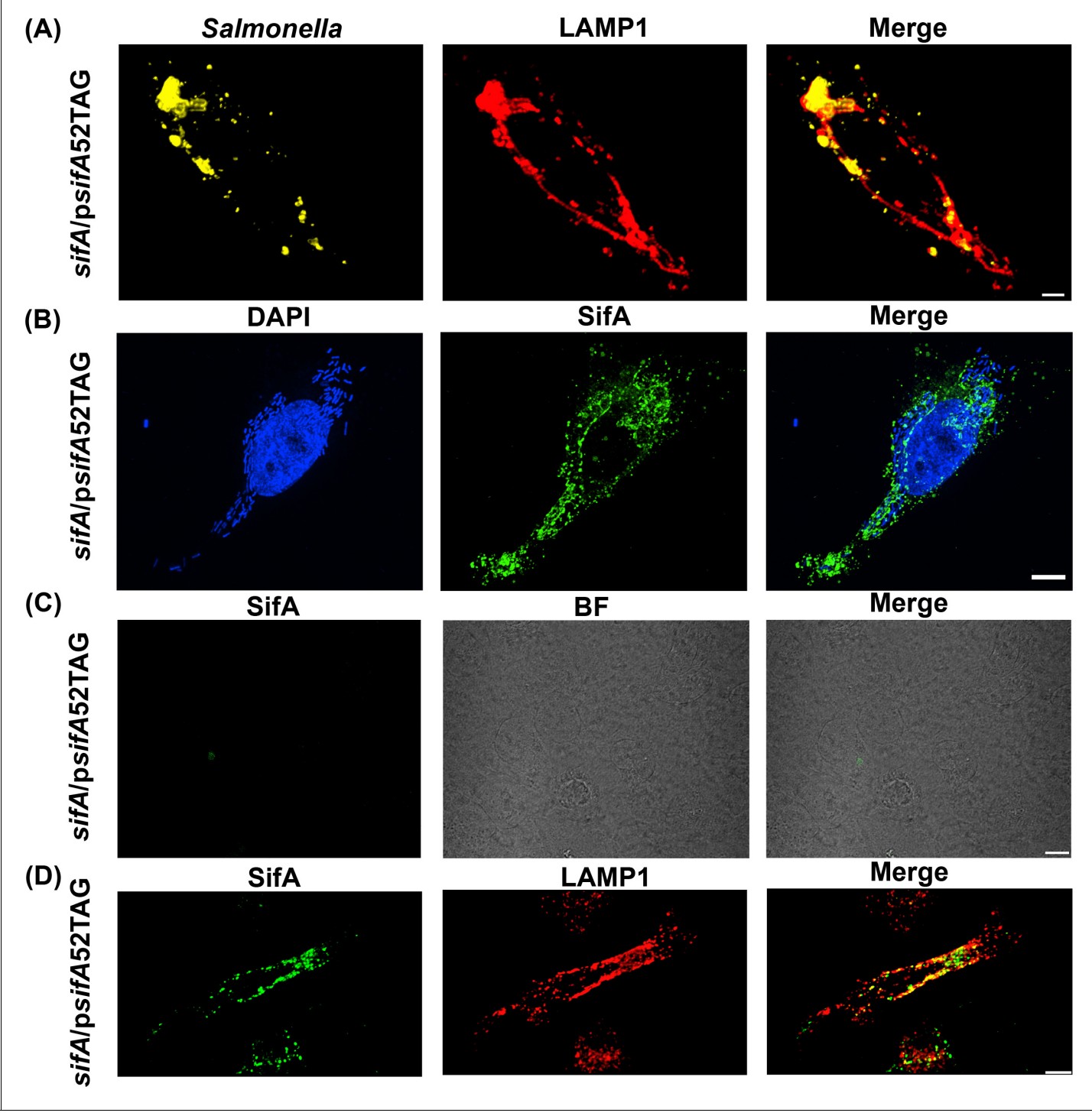

**Figure 3.** Functional complementation of SIF formation by SifA-E52AzF. (**A**) HeLa cells were infected with a *sifA* null strain of *Salmonella* expressing SifA-E52AzF and incubated for 16 hr. Infected cells were fixed with 4% PFA and immuno-stained for *Salmonella* (yellow) and LAMP1 (red). Images were acquired by spinning-disk SIM. Cells infected with *sifA* harboring p*sifA*52TAG bacteria were capable of forming SIFs that resembled those of wild-type infected cells. (**B**) SPAAC labeling of secreted SifA-E52AzF with BDP-DBCO. HeLa cells were infected with a *sifA* null strain expressing SifA-E52AzF in the presence of 1 mM AzF. AzF-tagged secreted SifA-E52AzF was labeled with BDP-DBCO (green). Host nuclei and *Salmonella* nucleoids were labeled with DAPI (blue) and images were acquired using spinning-disk SIM. (**C**) SifA-E52AzF labeling was absent in the absence of ncAA. HeLa cells were infected with the *sifA* null mutant of *Salmonella* expressing SifA-E52AzF in the absence of AzF for 12 hr. After 12 hr post infection, HeLa cells were incubated with 2.5 μM BDP-DBCO in DMEM with 10% FBS growth media for another 2 hr, followed by extensive washing to remove excess dye as described in Methods. At 16 hr post-infection, cells were fixed. SifA was absent in the infected cells that lacked AzF. Images were acquired by confocal

*Figure 3 continued on next page*

Figure 3 continued

microscopy. See *Figure 3—figure supplement 4* for statistical analysis for the average fraction of infected cells that contain LAMP1 positive SIFs and SIFs containing labeled SifA. (D) Colocalization of SifA with LAMP1. *Salmonella* secreted SifA-E52AzF in HeLa cells was labeled with BDP-DBCO (green). Cells were also immuno-stained for the endosomal membrane marker LAMP1 (red, middle panel). Images were acquired by confocal microscopy. From the merged image (right), it is evident that SifA is present within the SIFs. Statistical analysis is shown in *Figure 3—figure supplement 5*. The data are representative of at least three independent experiments. Scale bar, 10 μm (**A–D**).

The online version of this article includes the following source data and figure supplement(s) for figure 3:

**Figure supplement 1.** The amber suppression system (pEVOL) does not affect *Salmonella* pathogenesis.

**Figure supplement 1—source data 1.** Source data for *Figure 3—figure supplement 1A*.

**Figure supplement 1—source data 2.** Source data for *Figure 3—figure supplement 1B*.

**Figure supplement 2.** A *sifA* null strain lacks SIFs.

**Figure supplement 3.** No off-target labeling of secreted effectors is observed within host cells.

**Figure supplement 4.** The average fraction of infected cells that contain LAMP1 positive SIFs and SIFs containing labeled SifA.

**Figure supplement 4—source data 1.** Source data for *Figure 3—figure supplement 4*.

**Figure supplement 5.** The proportion of tubules with labeled SifA and its colocalization with LAMP1 (n = 27); n is the total number of cells analyzed.

**Figure supplement 5—source data 1.** Source data for *Figure 3—figure supplement 5*.

To further understand the localization of SifA with other components of the SIFs, we compared SifA and the secreted effector SseJ. We had previously demonstrated that a *Salmonella* strain expressing an SseJ-hemagglutinin (HA) fusion tag was competent for SPI-2-dependent SseJ secretion (*Chakraborty et al., 2015*; *Gao et al., 2018*); we used this strain to infect HeLa cells, and fixed and immunostained after 16 hr post-infection (*Figure 4*). In the super-resolution image, SseJ was readily observed as a mixture of regular, periodic clusters and as filamentous structures, as we reported (*Gao et al., 2018*). Secreted SifA-E52AzF was then labeled with BDP-DBCO in HeLa cells using the SPAAC reaction (*Figure 4A*). In vitro experiments had been unable to demonstrate an interaction with SifA and SseJ (*Ohlson et al., 2008*), although their opposing activities were essential for SIF formation (*Gao et al., 2018*; *Brumell et al., 2002*; *Kolodziejek et al., 2019*). Because of the close proximity of kinesin and SseJ observed previously (*Gao et al., 2018*), we hypothesized that SifA might also be co-localized with SseJ. We used spinning-disk SIM to examine the pattern of SifA and SseJ. They exhibited the same distribution pattern and appeared to be co-localized. In order to confirm their co-localization, we used modified Mander's colocalization analysis (*Manders et al., 1993*) for dense protein patterns (see Materials and methods) and obtained strong evidence for positive co-localization (*Figure 4Aiii*). In contrast, a mock infection established the background for cells lacking SseJ-SifA interactions (*Figure 4Aiv*).

SifA has been reported to interact with the motor protein kinesin-1 via the adapter protein SKIP (*Boucrot et al., 2005*). Therefore, kinesin and SifA should also be co-localized. We performed spinning-disk SIM imaging and investigated the pattern of SifA and kinesin (*Figure 4B*). SifA was in close proximity to kinesin. We again employed the modified Mander's co-localization analysis for dense protein patterns (*Manders et al., 1993*) and obtained additional evidence for positive co-localization (*Figure 4Biii*). A mock infection established a background level of SifA-kinesin interaction (*Figure 4Biv*). Since SseJ, SifA and kinesin were all co-localized, and SseJ was very sensitive to osmolality-induced pearling effects (*Gao et al., 2018*), these results confirmed that the motor protein kinesin exerts an indirect force on SseJ located on endosomal tubules via a SseJ-SifA-SKIP-kinesin protein complex. This is likely a weak interaction that can be easily disrupted by fixation (*Gao et al., 2018*).

## SifA does not continuously decorate filaments

Previous studies of *Salmonella* secreted effectors have shown that structural preservation of the SIFs was sensitive to fixation conditions (*Gao et al., 2018*). At high osmolality (4% PFA), secreted SseJ formed uniform, periodic clusters inside host cells, whereas in low osmolality fixation (0.2% GA), the same SseJ formed continuous filamentous structures. Because microtubules were disrupted by high osmolality fixation, it was proposed that the clusters of SseJ observed in the PFA-fixed cells were not representative of native structures (*Gao et al., 2018*). As a result, we questioned whether SifA in the SIFs formed continuous filamentous structures after fixation. Interestingly, we observed punctate structures of SifA, even during low osmolality fixation when microtubules were well-preserved

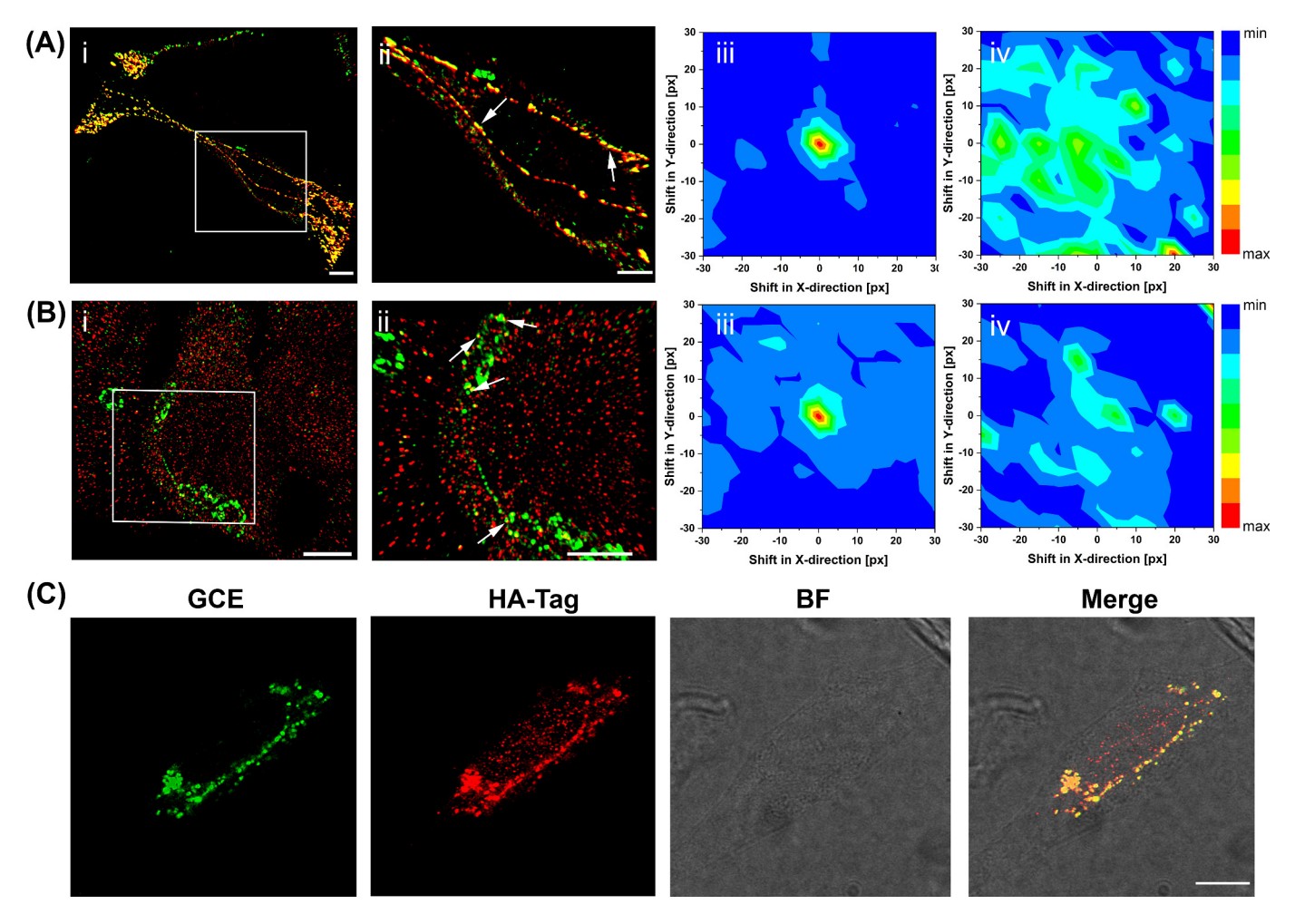

**Figure 4.** Super-resolution imaging of GCE-labeled SifA. (**Ai**) SifA colocalizes with SseJ. Spinning-disk SIM image of an infected HeLa cell, 16 hr after invasion, fixed with PFA and stained for SifA (green) and SseJ-HA (red). (**Aii**) Magnified view of SifA and SseJ in the boxed region at the far left (arrows indicate colocalization). (**iii**) Colocalization was analyzed by calculating Mander's M1 coefficient (SifA colocalizing with SseJ) for the original dual-color image (x = y = 0) and for control images generated by translating the spectral channels with respect to each other in x and y directions. This analysis generated a colocalization map, with a peak intensity at x = y = 0, indicating true colocalization (n = 3 cells). (**iv**) Colocalization analysis of mock-infected cells (n = 3 cells) stained for SifA-E52AzF and SseJ-HA showed no peak intensity at x = y = 0, indicating no colocalization of anti-HA antibodies and SifA. Bars, 10 μm (i), 5 μm (ii). (**B**) SifA colocalizes with kinesin. (**i**) Spinning-disk SIM image of an infected HeLa cell fixed with PFA at 16 hr post-infection and stained for kinesin (red) and SifA (green). (**ii**) Magnified view of SifA and kinesin in the boxed region (arrows indicate colocalization). (iii and iv) Colocalization analysis was performed as described in (**A**). The same color map scaling was used for comparison. px, pixels. See *Figure 4—figure supplement 1* for statistical analysis. Scale bar, 5 μm (i), 2 μm (ii). (**C**) Bioorthogonal fluorescence imaging of SseJ-Y10TCO-HA with tetrazine fluorophores versus immunofluorescence staining. HeLa cells expressing SseJ-Y10TCO-HA in the presence of TCO*Lys were labeled with BDP-Tz under physiological conditions, briefly washed, and subjected to anti-HA immunofluorescence staining. Images were acquired by confocal microscopy. The data are representative of at least three independent experiments. Scale bar, 10 μm.

The online version of this article includes the following source data and figure supplement(s) for figure 4:

**Source data 1.** Source data for *Figure 4(iii-iv)*..
**Source data 2.** Source data for *Figure 4(iii-iv)*.
**Figure supplement 1.** The proportion of tubules with labeled SifA and its colocalization with SseJ (n = 33) and kinesin (n = 35); n is the total number of cells analyzed.
**Figure supplement 1—source data 1.** Source data for *Figure 4—figure supplement 1*.
**Figure supplement 2.** GA fixation preserves microtubules.
**Figure supplement 3.** Functional complementation of SIFs by expression of fluorescent SifA-E52Hco.

(*Figure 4—figure supplement 2*). Imaging SifA labeled with BDP-DBCO in live cells prior to fixation further confirmed that, unlike SseJ, SifA does not form continuous filamentous structures within the SIFs (*Supplementary file 1*). Thus, the stoichiometry of SifA to SseJ cannot be 1:1.

## Genetically encoded fluorescent ncAA enables direct visualization of secreted SifA in host cells

The specificity and efficiency of intracellular protein labeling using SPAAC is influenced by the cellular properties of the particular fluorophore. Alternatively, one can apply GCE to introduce a relatively compact fluorescent ncAA (fncAA) site-specifically into the target effector protein SifA. We labeled SifA with the fncAA, 7-(Hydroxy-coumarin-4-yl) ethylglycine (Hco) by first infecting HeLa cells with the Δ*sifA* strain expressing SifA-E52Hco. After 16 hr post-infection, cells were fixed and immuno-stained with anti-LAMP1 antibody to reveal the SIFs. SIFs were also rescued by the expression of fully functional SifA-E52Hco using GCE (*Figure 4—figure supplement 3*). Secreted SifA in host cells was visualized using two-photon microscopy. In the figure (*Figure 4—figure supplement 3B*), it can be seen that SIFs were formed by secreted SifA in which Hco was incorporated site-specifically. Thus, without any click reactions, site-specific labeling of SifA with Hco directly localizes secreted effectors in host cells in vivo. This result further validates that SifA co-localizes with SseJ, and the presence of SifA within the SIFs.

## GCE-labeling is superior to small protein tags and antibody staining

GCE-based bio-orthogonal labeling offers an effective way for direct labeling using fluorescent dyes of secreted effector proteins inside host cells. To directly compare the image quality of a secreted effector in HeLa cells, we compared SseJ that was labeled via a C-terminal HA epitope fusion with GCE-labeled SseJ, where an amber stop codon was introduced at the 10th position (p*sseJ*-10TAG-HA). We then infected HeLa cells with *Salmonella* transformed with either p*sseJ*-10TAG-HA, or secreted SseJ-F10TCO was labeled with BDP-Tz via the SPIEDAC reaction (*Agard et al., 2004*; *Plass et al., 2012*). After labeling, excess dye was washed out and cells were fixed and immuno-stained with anti-HA antibody to visualize the secreted SseJ. SseJ-F10TCO was secreted and labeled with BDP-Tz dye (*Figure 4C*, green). SseJ labeled via GCE (green) and HA (red) was compared side by side and also in the merged image. Clearly, they are co-localized, but the reduced background in the GCE image is readily apparent.

## SsaP functions to control SPI-2 type III secretion

T3SSs switch from secreting apparatus components to effectors in a conserved process referred to as the substrate specificity switch. This step is crucial to decide the timing of effector secretion. In SPI-2, SsaP is predicted to be the substrate specificity switch, based on homology to other T3SSs as well as the location of *ssaP* relative to its homologue in the SPI-1 T3SS. To examine the role of SsaP in SPI-2 secretion, we created an *ssaP* deletion mutant, and then infected HeLa cells with the *ssaP* mutant that contained p*sseJ*-HA. After 16 hr post-infection, cells were fixed and immuno-stained with anti-LAMP1 and anti-HA antibodies to reveal the secreted SseJ and the SIFs. We observed that the *ssaP* mutant was defective in SPI-2 secretion (*Yu et al., 2018*), as evident by the lack of SseJ secretion, and the absence of SIFs (*Figure 5—figure supplement 1*). As expected, the *ssaP* null mutant was unable to survive in HeLa cells (*Figure 5—figure supplement 2*), further evidence that SsaP was required for *Salmonella* virulence.

The outer pore of the injectisome is the translocon, SseB, and we can observe its presence by fluorescence immuno-staining (*Chakraborty et al., 2015*), or by western blot (*Figure 2—figure supplement 1*). Compared to the wild-type, the *ssaP* deletion mutant still expressed *sseB*, but it was no longer secreted. It was then of interest to examine the cellular localization of SsaP to determine whether it was localized to the T3SS basal body, which was shown to be mostly confined to the cell poles (*Chakravortty et al., 2005*;*Chakraborty et al., 2015*). We constructed N-terminal and C-terminal photoactivatable mCherry (PAmCherry) fusions in the *ssaP* null strain, placing the fusions at the lambda attachment site as previously (*Liew et al., 2019*). We then examined the ability of these fusions to support translocon secretion (*Figure 2—figure supplement 1*). A fusion of PAmCherry to the C-terminus of SsaP supported secretion of SseB translocons, but an N-terminal fusion did not. Furthermore, western blot analysis revealed that the C-terminal PAmCherry fusion was cleaved, thus

preventing us from imaging SsaP (*Figure 2—figure supplement 1B*). To overcome this problem, we constructed six different SsaP-HA fusions, sandwiching the HA tag in likely internal loops of SsaP (*ʌssaP*-1-HA, *ʌssaP*-20-HA, *ʌssaP*-55-HA, *ʌssaP*-86-HA, *ʌssaP*-110-HA, and *ʌssaP*-124-HA). Three of the fusions were functional and restored SseB secretion (*Figure 2—figure supplement 3*).

Western blot analysis also suggested that SsaP-HA was secreted under SPI-2 inducing conditions in vitro (*Figure 2—figure supplement 3*). To examine whether SsaP was also secreted during infection, we infected HeLa cells with *Salmonella* containing the HA-tagged SsaP constructs (*ʌssaP*-55HA and *ʌssaP*-110HA) and immunostained with anti-HA antibody (*Figure 5—figure supplement 3*). Although SsaP was expressed at very low levels, there was evidence of its secretion into the HeLa cell cytosol. In order to verify this result, we labeled SsaP using GCE with the fluorogenic dye, SiR-Tz via the SPIEDAC click reaction, to minimize background signals. We determined that TCO*Lys-incorporated SsaP was functional by infecting HeLa cells with the Δ*ssaP* strain complemented with p*ssaP*65TAG-*sseJ*-HA. At 16 hr post-infection, cells were fixed with PFA and immuno-stained with anti-LAMP1 and anti-HA antibodies to reveal the SIFs and secreted SseJ. SseJ was clearly secreted, and SIFs were formed (*Figure 5—figure supplement 1B–C*). These observations confirmed that engineered SsaP rescued SPI-2 secretion (SseJ) and was functional, whereas an *ssaP* null strain failed to secrete SPI-2 effectors (*Figure 5—figure supplement 1A*).

Once we established that GCE-labeled SsaP was functional, the next step was to determine whether it was secreted during infection. SsaP-Y65TCO was labeled with SiR-Tz via the SPIEDAC reaction after infection of HeLa cells. Cells were fixed and immuno-stained as before and directly analyzed using spinning-disk SIM (*Figure 5A*). Intracellular fluorescent puncta were clearly observed in the SiR-Tz channel when SsaP-Y65TCO was expressed in the presence of TCO. When HeLa cells were infected with *Salmonella* containing wild-type *ssaP* (i.e. lacking the TAG codon) and TCO*Lys was added, non-specific fluorescence signals in the SiR-Tz channel were absent (*Figure 5B*), further validating the specificity, labeling and imaging in host cells of even the low-abundant SsaP. At earlier infection times (8 hr post infection), SsaP was mostly associated with bacteria (*Figure 5—figure supplement 4*), whereas at later times (16 hr), SsaP was secreted (*Figure 5Ai*, inset). As we observed with the SPI-2 injectisome (*Liew et al., 2019*; *Chakraborty et al., 2015*; *Chakravortty et al., 2005*), SsaP mostly localized to the poles, although not every pole where SsaP was located had an injectisome (*Figure 5—figure supplement 4C*). Thus, using GCE, we established that SsaP was secreted and we could visualize secreted SsaP within the SIFs for the first time.

## Discussion

Intracellular pathogens such as *Salmonella*, *Shigella*, and *Yersinia* use T3SSs to deliver their virulence factors into the host. Extracellular pathogens (including: EPEC, EHEC, *Pseudomonas aeruginosa*) also employ these complex nanomachines to inject their virulence factors. The revolution in super-resolution imaging methods have enabled the visualization of virulence factors at unprecedented resolution (*Singh and Kenney, 2021*). However, certain limitations in labeling have prevented a more thorough analysis, as photoactivatable fusion proteins are unable to be translocated through the T3SS pore; the inherent ability of some fusion proteins to cluster can result in clustering artefacts and spurious analysis, and occasionally, the fusion protein is cleaved, as we observed with PAm-Cherry-SsaP. GCE circumvents all of these limitations, enabling the site-specific labeling of proteins of interest. It also provides an opportunity to visualize proteins that are not abundant, such as SsaP, through the use of newer, brighter fluorophores. Recent advances in understanding the context around the engineered TAG codon (*Xu et al., 2016*) also provides a rapid and relatively low risk ability to label any number of proteins. Thus, the approach, as we described herein will be widely accessible for researchers to visualize proteins of interest in pathogens, commensal bacteria, as well as eukaryotic hosts.

We used GCE to study SifA, a critical SPI-2 effector protein that modulates membrane trafficking inside host cells and SsaP, reported to be the substrate specificity switch. Both proteins had been refractive to labeling by other methods. Selective and site-specific labeling of SifA was achieved by genetic incorporation of bioorthogonal ncAAs (AzF and Hco) via GCE technology utilizing an orthogonal aminoacyl-tRNA synthetase/tRNA pairs, and an alternative codon (UAG, amber stop codon). Treatment of AzF-labeled SifA (SifA-E52AzF) with DBCO containing a fluorophore (BDP-DBCO) provided an alternative protein labeling strategy and enabled visualization of translocated SifA effector

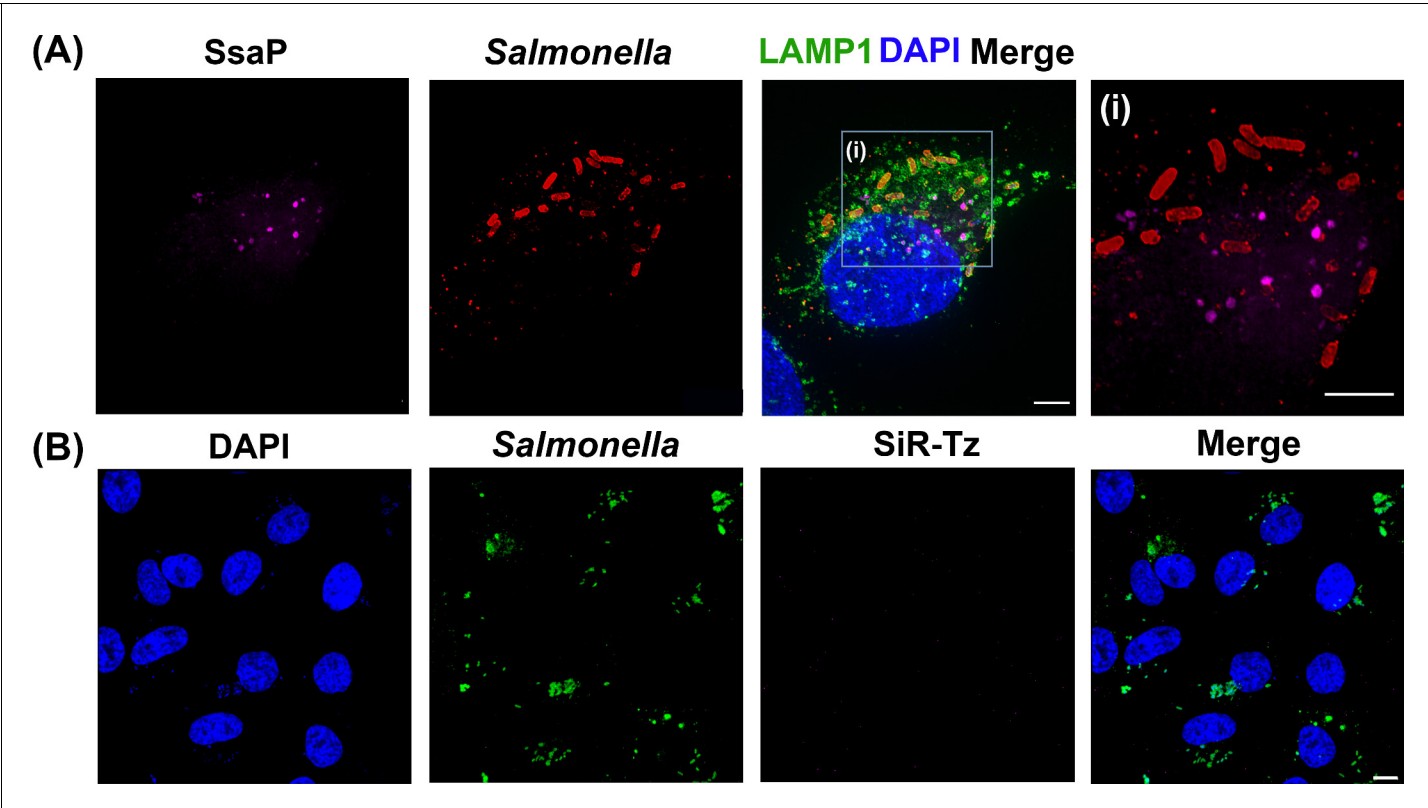

**Figure 5.** SsaP is secreted during HeLa cell infection. (**A**) SPIEDAC labeling of secreted SifA-Y65TCO with fluorogenic SiR-Tz. HeLa cells were infected with the *ssaP* null mutant of *Salmonella* expressing SsaP-Y65TCO in the presence of TCO*Lys for 12 hr. After 12 hr post infection, HeLa cells were incubated with 1.5 μM SiR-Tz in DMEM with 10% FBS growth media for another 2 hr, followed by extensive wash out of excess dye with fresh growth media as described in Materials and methods. At 16 hr post-infection, cells were fixed. Cells were also immuno-stained for the endosomal membrane marker LAMP1 (green), LPS (Red) and DAPI (Blue) and imaged by spinning-disk SIM. From the merged image (right), it is evident that SsaP is present within the LAMP1-positive endosomes. A higher magnification spinning-disk SIM image of a SIF-positive HeLa cell from the boxed region (i) is shown on the right-most panel without the green channel displayed (i). It clearly shows secreted SsaP from *Salmonella* (boxed region). (**B**) SsaP was absent in infected cells that lacked *ssaP*. HeLa cells that were infected with an *ssaP* null mutant in the presence of TOC*Lys lack an SsaP signal. Note the absence of SIFs in the merged image (right) in the *ssaP* null strain. The data are representative of at least three independent experiments. Scale bar, 5 μm (**A** and i) and 10 μm (**B**).

The online version of this article includes the following figure supplement(s) for figure 5:

**Figure supplement 1.** An *ssaP* null mutant is defective in SPI-2 secretion.

**Figure supplement 2.** Survival of *S*. Typhimurium SsaP-55HA mutant in HeLa cells.

**Figure supplement 3.** SsaP is secreted.

**Figure supplement 4.** SsaP was associated with bacteria and mostly localized to the poles during early infection times.

**Figure supplement 5.** Schematic of endosomal tubule (SIF) formation.

proteins over many hours after infection of the host. The ability of SifA-E52AzF to induce LAMP1-positive SIFs demonstrates that its function was not impaired by the ncAA tagging strategy and established the presence of SifA within the SIFs (*Figure 3*). What was lacking in previous studies (*Ohlson et al., 2008*; *Gao et al., 2018*; *Rajashekar et al., 2008*) was an understanding of where SifA was located with respect to the kinesin motor and the effector SseJ. GCE has overcome these limitations of SifA labeling, allowing its visualization for the first time after secretion by the T3SS, and with native levels of protein expression (*Figure 3* and *4*). Previous studies have shown that SifA interacts with kinesin through SKIP (*Boucrot et al., 2005*). In a *sifA* deletion strain, SIFs were absent (*Figure 3—figure supplement 2*) and SseJ remained associated with the SCV (*Gao et al., 2018*). Together with our observation that SifA colocalized with SseJ (*Figure 4A*) and kinesin (*Figure 4B*), we established that a SseJ-SifA-SKIP-kinesin protein complex is required for endosomal tubulation. In the case of SsaP, we were able to visualize it inside bacterial cells and discovered that after

infection of HeLa cells by *Salmonella,* SsaP was secreted. SsaP was mostly localized to the poles, although the presence of SsaP did not ensure the presence of an injectisome at the same location.

The numerous challenges of using GCE include the selection of fluorophores, which should be carefully selected on the basis of their chemical properties, such as membrane permeability, retention and distribution, which all can affect labeling efficiency. Many of these challenges involved in the generation of fluorescently labeled effector proteins can be overcome by genetically encoding fluorescent amino acids directly. Using our effector protein labeling strategy (*Figure 1*), we also successfully demonstrated that SIFs containing SifA can be visualized by genetically encoding a single fluorescent amino acid (Hco) without the need for additional click chemical labeling procedures (*Figure 4—figure supplement 3*).

Earlier analysis of SseJ clusters provided evidence of a pearling effect, because endosomal tubulation is a force-driven, osmotically sensitive process. The pearling transition or instability is a sequential beading of tubular vesicles first observed in preparations of endoplasmic reticulum membranes that were shown to be sensitive to hypotonic shock (*Dabora and Sheetz, 1988*). Pearling can also be driven by high osmolality (*Gao et al., 2018*; *Yanagisawa et al., 2008*; *Pullarkat et al., 2006*; *Derényi et al., 2002*; *Sanborn et al., 2013*), as we observed previously (*Gao et al., 2018*). Kinesin was identified as the motor protein that supplies the force for tubulation, via its interaction with an SseJ-SifA-SKIP tubulating complex (*Figure 4*). The observation that SifA was not subject to pearling effects also implies that it is not directly force-dependent. These results lead to a model for membrane tubulation, as depicted in *Figure 5—figure supplement 5*. When *Salmonella* resides within the SCV, the sensor kinase EnvZ senses the acidic vacuolar environment and drives OmpR-dependent acidification of the bacterial cytoplasm to pH 5.6 (*Chakraborty et al., 2015*). Acidification triggers an up-regulation of the SsrA/B two-component system located on SPI-2. DNA binding by SsrB is acid-sensitive and increases by >50%, driving the expression of the SPI-2 T3SS structural genes and effectors (*Liew et al., 2019*; *Kenney, 2019*). The kinesin motor interacts with an SseJ-SifA-SKIP protein complex, pulling the vacuolar membrane along microtubules (*Figure 5—figure supplement 5*) and incorporating LAMP1-positive endosomes into the SIFs. Tubulation is a force-driven process through the concerted action of SseJ-SifA-SKIP-kinesin protein complexes, where force is provided by the motor protein kinesin.

In summary, we report that genetically encoded unnatural amino acids provide an alternative route to effector protein labeling methodology when conventional methods fail. In addition, we demonstrate the superior labeling and reduced background using GCE compared to immuno-staining (*Figure 4C*). We successfully visualized SifA by genetically encoding non-fluorescent, or fluorescent ncAAs without compromising its biological function. While some restrictions such as low ncAA incorporation efficiency and the photophysical limitations of the fluorophore might apply, it is evident that the approach is efficient enough to visualize and localize a secreted effector in host cells, even when it is not very abundant, as was the case with SsaP. Considering the simplicity of the approach and the bioconjugation reaction, we anticipate that straightforward extensions of the technology should provide new and exciting opportunities for imaging and investigation of other effector proteins during infection by bacterial pathogens.

## Materials and methods

### Reagents

4-Azido-L-phenylalanine (AzF), trans-Cyclooct-2-en – L - Lysine (TCO*Lys) and 7-(Hydroxy-coumarin-4-yl) ethylglycine (Hco) were obtained from MedChemExpress, SiChem GmbH and Bachem (Switzerland), respectively. BDP FL DBCO alkyne, SiR-tetrazine (SiR-Tz), and BDP-tetrazine (BDP-Tz) were purchased respectively from BroadPharm (San Diego, USA), Spirochrom (Switzerland), and Lumiprobe (USA). All the chemicals were purchased from Sigma-Aldrich unless otherwise noted. Chemicals and oligonucleotides obtained from commercial sources were used without further purification.

### Plasmids and strains

*Salmonella enterica* serovar Typhimurium strain NCTC12023 was the wild-type strain, and various isogenic mutants are described in the text. The plasmid pEVOL, encoding an AzF-specific engineered pair of tyrosyl-tRNA synthetase/amber suppressor tRNA derived from *Methanococcus*

*jannaschii* (Plasmid ID: 31186), was obtained from Addgene and used without modification. For complementation, a *sifA* null strain was used harboring p*sifA*52TAG (copy number ~5) and pEVOL-pAzF or pEVOL-CouRS. Bacteria were grown overnight in LB broth at 37°C with shaking. The *sifA* gene along with its native promoter was amplified from *Salmonella* genomic DNA. An amber codon (TAG) was introduced at position E52 of SifA by site-directed mutagenesis. TAG-containing PCR fragments were subcloned into pWSK29 between SmaI and EcoRI restriction sites. Similarly, for the complementation of SsaP, a *ssaP* null strain harboring p*ssaP*65TAG and pEVOL-PylRS-AF (a *Methanosarcina mazei* tRNAPyl$_{CUA}$/PylRS$^{AF}$ pair encoded in the plasmid pEVOL) was used. TAG was introduced at Y65 of SsaP ensuring a higher likelihood of labeling with AAT-TAG-ACT motif (*Lammers et al., 2010*). The *sseJ* gene along with its native promoter was amplified from *Salmonella* genomic DNA. An amber codon (TAG) was introduced at position Y10 of SseJ by site-directed mutagenesis ensuring the AAT-TAG-ACT motif (*Xu et al., 2016*). TAG-containing PCR fragments were subcloned into pWSK29 between XbaI and SacI restriction sites. Wilde-type *Salmonella* strain was used to harbor p*sseJ*10TAG and pEVOL-PylRS-AF. The identity of each constructs was verified by sequencing. Chromosomal fusions of *ssaP* with the hemagglutinin (HA) tag and PAmCherry were constructed as described previously (*Liew et al., 2019*). pEVOL-pAzF was a gift from Peter Schultz (Addgene plasmid # 31186). pEVOL-CouRS was generously provided by Prof. Hyun Soo Lee at Sogang University, Seoul, Korea. pEVOL-PylRS-AF was generously provided by Prof. Edward Lemke at European Molecular Biology Laboratory (EMBL), Heidelberg, Germany. pEVOL containing Δ*sifA* 12023 or Δ*ssaP* or wild-type *Salmonella* were obtained by transformation with pEVOL, followed by incubation at 37°C on LB plates in the presence of 25 µg/mL chloramphenicol. The pEVOL-containing strains were subsequently transformed with either p*sifA*52TAG or p*ssaP*65TAG or p*sseJ*10TAG expression construct and incubated at 37°C on LB plates in the presence of 25 µg/mL chloramphenicol and 100 µg/mL ampicillin.

## Expression of ncAA-bearing proteins in *Salmonella*

Overnight bacterial cultures of Δ*sifA* 12023 harboring p*sifA*52TAG expression constructs and plasmid pEVOL-pAzF or pEVOL-CouRS were grown at 37°C in LB medium until OD$_{600}$ reached 0.6, at which point LB media was replaced with MgM pH 5.6 media containing 1 mM AzF or 1 mM Hco. Bacteria were continuously grown at 34°C for another 30 min before induction in the presence of 0.2% arabinose, 25 µg/mL chloramphenicol and 100 µg/mL ampicillin, for 8 hr. Similarly, for in-vitro the expression of SsaP in MgM (pH 5.6), Δ*ssaP* harboring p*ssaP*65TAG expression constructs and pEVOL-PylRS-AF was used along with 1 mM TCO*Lys.

## Mammalian cell culture

HeLa cells, human epithelial cell line (American Type Culture Collection; ATCC CCL-2) were used for *Salmonella* infections. The HeLa cell line was authenticated by ATCC (STR analysis) and the absence of mycoplasma contamination was confirmed using Hoechst DNA stain (indirect), agar culture (direct), a PCR-based assay and Lonza's MycoAlert PLUS Mycoplasma detection kit. HeLa cells were maintained in high-glucose (4.5 g/L) Dulbecco's modified Eagle's medium (DMEM) containing glutamine and pyruvate (Thermo Fisher) and supplemented with 10% inactivated fetal bovine serum (FBS; Thermo Fisher) at 37°C and antibiotics (antibiotic-antimycotic final 5x stock; Thermo Fisher) in a humidified 5% CO$_2$ incubator. Cells were gently trypsinized with Accutase (StemPro Accutase; ThermoFisher), and ~5×10$^4$ cells were seeded into 24-well plates with 12 mm sterile coverslips and grown 24 hr before infection.

## HeLa cell infection and labeling of SifA, SsaP, and SseJ

Δ*sifA* strains harboring p*sifA*52TAG and pEVOL-pAzF or pEVOL-CouRS were cultured overnight in standard LB broth in the presence of antibiotics. Overnight cultures were subcultured 1:30 in 3 mL LB for 5 to 6 hr OD$_{600}$, ~2–3. Cultures were then diluted to an OD$_{600}$ = 0.2. HeLa cells were infected with a multiplicity of infection (MOI) of 100:1 (bacteria/HeLa cell) and bacteria were internalized for 30 min. Extracellular bacteria remaining were removed by washing three times in Dulbecco's phosphate-buffered saline (DPBS), followed by incubation with 100 µg/mL of gentamicin for 1 hr in DMEM with 10% FBS. The cells were again washed three times with DPBS and further incubated in DMEM with 10% FBS, 0.2% arabinose, 0.5 mM p-azido-L-phenylalanine or 0.25 mM 7-(Hydroxy-

coumarin-4-yl) ethylglycine, and gentamicin (20 µg/mL) for the rest of the infection. After 10 hr post infection, media was exchanged with normal growth media without ncAAs and washed four times with fresh DMEM media. Similarly, for the labeling of secreted SsaP or SseJ, HeLa cells were infected with ΔssaP strains (harboring pssaP65TAG and pEVOL-PylRS-AF) or wild-type *Salmonella* (harboring psseJ10TAG-HA and pEVOL-PylRS-AF) in presence of 0.25 mM TCO*Lys. As a control, similar infections of HeLa cells were also carried out in the absence of unnatural amino acids. For fluorescent labeling of AzF-tagged SifA, TCO-tagged SsaP and SseJ in HeLa cells, 2.5 µM BDP-DBCO, 1.5 µM SiR-Tz, BDP-Tz in DMEM with 10% FBS growth media was added respectively to each well, and after 1 hr, cells were washed 4 × 30 min with fresh growth media. Cells were fixed for imaging at 16 hr post-infection as described below.

## Immunostaining and confocal microscopy

Adherent HeLa cells were infected as described above and were fixed with either using 200 µl of methanol-free 4% PFA (EM grade; TaKaRa) in PBS for 10 min or 0.2% GA, washed thrice with PBS, and resuspended in 300 µl PBS. After several washes with PBS, cells were processed for immunostaining. Fixed HeLa cells were incubated with primary antibody solution (a PBS-based buffer containing 5% bovine serum albumin and 0.1% saponin). For secreted SseJ staining, rabbit anti-HA primary antibody (Sigma; H6908) was used (rabbit, 1:500 dilution). Rabbit monoclonal anti-KIF5B (Abcam catalog no. ab167429) was used for kinesin, and a monoclonal mouse anti-LPS (1:300 dilution) was used for *Salmonella* LPS. For LAMP1, a monoclonal mouse anti-LAMP1 primary antibody (Santa Cruz Biotechnology; clone H4A3, 0.4 g/mL) was used. After 1 hr, the primary antibody solution was removed, and cells were gently washed three times with PBS containing 0.1% Tween-20. Secondary antibody (goat anti-mouse 647 [Thermo Fisher, catalog no. A-21237], donkey anti-rabbit 555 [Invitrogen, catalog no. A-31572], goat anti-mouse 488 [Thermo Fisher, catalog no. A-28175], and donkey anti-rabbit 488 [catalog no. A-21206], 2 g/mL] diluted at 1:500 in PBS-BSA-saponin buffer was subsequently added for 1 hr at room temperature in the dark. After three washing steps DAPI, 600 nM in PBS was applied for 10 min to stain HeLa cell and *Salmonella* DNA. The cells were stored in PBS in the dark at 4°C. Confocal imaging was performed on an inverted Nikon A1R Module equipped with CFI Plan Apochromat VC 100XH, 1.4 numerical aperture objective lens using 405, 488, 561, and 647 nm Coherent lasers. Filter sets were selected to match the excitation and emission properties of the fluorophores. Infections were performed tens of times and hundreds of cells were imaged per infection, and the figures display representative images.

Confocal imaging of SsaP-labeled *Salmonella* was performed on a Leica TCS626 SP8 3X system (Leica Microsystems, Wetzlar Germany) equipped with a Leica HC PL 627 APO CS2 100×/1.4 oil immersion objective. A tunable (470–670 nm) pulsed white light laser 628 (Leica WLL) was used for excitation. Images were processed with the image analysis software, Imaris Pro 9.3.0. GraphPad Prism nine for MacOS, GraphPad Software, La Jolla California USA, www.graphpad.com was used to make all the graphs and statistical analysis.

## Immunoblotting

Overnight cultures of wild-type *Salmonella*, ssaP, ΔssaP, ΔssaP-PAmCherry, ΔPAmCherry-ssaP, ΔssaP-1-HA, ΔssaP-20-HA, ΔssaP-55-HA, ΔssaP-86-HA, and ΔssaP-110-HA mutant strains were inoculated in LB for 12 hr. Cells were washed with LB-MES (pH 5.6) and inoculated in LB-MES (pH 5.6) for 8 hr. Cells were removed from the culture by centrifugation (6000 x g, 20 min, 4°C), and the supernatant was filtered through a 0.22 µm pore size binding filter (Millipore, Billerica, MA). From the supernatant, the secreted protein fraction was isolated by 10% trichloroacetic acid precipitation and washed three times with −20°C acetone, and then air dried. For whole cell fraction, cells were lysed in lysis buffer (0.1M NaCl, 0.05M Tris-HCl, pH 7.2). Total bacterial cells and precipitated secreted proteins were dissolved in SDS-PAGE loading buffer and separated by 12% SDS-PAGE, transferred to PVDF membrane (Millipore). The membrane was incubated with anti-SseB (1:5000) or anti-GroEL (1:5000) antibodies in PBS buffer with (0.05% Tween 20) followed by anti-rabbit secondary antibody (1:10,000, Santa Cruz Biotechnology).

For SsaP-HA detection, blots were incubated with a primary antibody directed against the HA epitope tag (1:1000) and a secondary antibody anti-mouse IgG antibody (1:10,000, Santa Cruz Biotechnology) conjugated to horseradish peroxidase (HRP). Secondary antibody was detected by

SuperSignal Pico PLUS Chemiluminescent substrate (Thermofischer Scientific). For SsaP-PAmCherry, a polyclonal anti-mCherry primary antibody raised in rabbits (1:1000, abcam) was then added and incubated for 1 hr. After the incubation, the membrane was washed three times, and the primary antibody was detected with anti-rabbit immunoglobulin raised in goat. Blots were visualized with a Chemidoc imaging system (Bio-Rad).

## Structured-illumination microscopy (SIM)

SIM imaging was performed on a CSU-W1 spinning disk confocal system attached to a Nikon Eclipse Ti-E inverted microscope combined with the Live-SR system (Roper Scientific), controlled by Meta-Morph software (Molecular device), supplemented with a 100X oil 1.45 NA CFI Plan Apo Lambda oil immersion objective and sCMOS camera (Prime 95B, Photometrics). Excitation and band-pass filters were as follows: BDP DBCO dye was imaged using 488 nm excitation and emission of 525/30 nm, and Alexa Fluor 647 was imaged using 642 nm excitation and emission of 705/72 nm. The raw images were processed to superresolution images by using the Live-SR algorithm reported by *York et al., 2013*.

## Two-photon imaging of SifA labeled with fluorescent ncAA (Hco) in HeLa Cell

Fluorescent images using two-photon microscopy were acquired using a Leica TCS SP5 X confocal microscope equipped with a Leica HC PL APO 100x/1.40 Oil immersion objective. Illumination was provided by a Ti-Sapphire IR laser (Chameleon Ultra II). The cells were excited at 730 nm (two-photon) to acquire Hco fluorescence images at 420–500 nm emission wavelength. The imaging setup was controlled by the Leica Application Suite-Advanced Fluorescence (LAS AF). Images were processed using Imaris 9.3.0.

## Colocalization analysis

Colocalization between SifA and SseJ-HA were determined from Spinning-disk SIM images using a variation of Mander's colocalization analysis (*Gao et al., 2018*; *Manders et al., 1993*) and Fiji (*Schindelin et al., 2012*). Using Otsu's algorithm, maximum intensity projections of both spectral channels were first converted into a binary image. Co-localizing pixels were identified by multiplying the binary images. In order to segregate true co-localization and random overlap, one image channel (SseJ) was translated in x, y using a custom-written macro in Fiji (*Gao et al., 2018*). For each translation step, the M1 (SseJ co-localizing with SifA) and M2 (SifA co-localizing with SseJ) values of the two images were calculated. Co-localization maps were created using OriginPro 2017. One to three regions of interest (ROIs) per cell (3 cells) were analyzed. The M1 value for each ROI was normalized (division by maximum value), followed by averaging of all M1 values as described previously (*Gao et al., 2018*). Similar colocalization analysis was also carried between SifA and Kinesin.

## Acknowledgements

This work was supported by a Texas STAR award and start-up funds from the University of Texas Medical branch, Galveston, TX to LJK. We thank Prof. Peter Schultz (Scripps Research Institute) for providing plasmid pEVOL-pAzF, Prof. Hyun Soo Lee (Sogang University, Seoul, Korea) for plasmid pEVOL-CouRS, Prof. Edward Lemke (European Molecular Biology Laboratory, Heidelberg, Germany) for plasmid pEVOL-PylRS-AF. We also thank Prof. Mike Heilemann and Dr. Christoph Spahn (Goethe University, Frankfurt, Germany) for the customized code for colocalization analysis and valuable comments on the work. We are extremely grateful for the superb expertise of the Mechanobiology Microscopy Core Facility, Science Communication Core and the Wet Lab Core.

## Additional information

### Funding

| Funder | Grant reference number | Author |
|---|---|---|
| Ministry of Education - Singapore | MOE2018-T2-1-038 | Linda J Kenney |

| | | |
|---|---|---|
| University of Texas Medical Branch | Texas STAR award | Linda J Kenney |

The funders had no role in study design, data collection and interpretation, or the decision to submit the work for publication.

### Author contributions
Moirangthem Kiran Singh, Designed the research, performed the experiments, analysed the data, wrote the paper; Parisa Zangoui, Yuki Yamanaka, performed the experiments; Linda J Kenney, designed the research, wrote the paper

### Author ORCIDs
Moirangthem Kiran Singh https://orcid.org/0000-0001-9620-5454
Parisa Zangoui https://orcid.org/0000-0003-2875-6726
Linda J Kenney https://orcid.org/0000-0002-8658-0717

### Decision letter and Author response
Decision letter https://doi.org/10.7554/eLife.67789.sa1
Author response https://doi.org/10.7554/eLife.67789.sa2

## Additional files

### Supplementary files
• Supplementary file 1. Supplemental file for Figure 4: Live cell imaging of secreted SifA labelled with BDP-DBCO dye during HeLa cell infection.

• Transparent reporting form

### Data availability
All data generated or analysed during this study are included in the manuscript and supporting files. Source data of Figure 4A-B, Figure 2-figure supplement 5, Figure 3-figure supplement 1, Figure 3-figure supplement 4, Figure 3-figure supplement 5, Figure 4-figure supplement 1 are included.

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
