## [Decision Letter]

**Acceptance summary:**

A tour-de-force, this study applies genomic expansion mediated protein labeling to investigate the mechanism of *Salmonella*-induced filament (SIF) formation in infected cells as catalyzed by secreted effectors of the *Salmonella* pathogenicity island 2 type 3 secretion system. Using this cutting edge approach, the authors demonstrate that a key factor, SsaP, controls the secretion-specificity switch from injectisome substrate secretion to translocon and effector secretion, clarifying the mechanism driving SIF formation.

**Decision letter after peer review:**

[Editors’ note: the authors submitted for reconsideration following the decision after peer review. What follows is the decision letter after the first round of review.]

Thank you for submitting your article "Labeling and visualization of *Salmonella* secreted effector SifA by genetic code expansion" for consideration by *eLife*. Your article has been reviewed by 3 peer reviewers, and the evaluation has been overseen by Petra Levin as the Reviewing Editor and Gisela Storz as the Senior Editor. We are sorry to be the bearer of unpleasant news, but we regret that we cannot recommend the study for publication in *eLife* at this time.

The Reviewing Editor has highlighted the concerns of all three reviewers as well as Dr. Storz. We have included the separate reviews below for your consideration. If you have any questions, please do not hesitate to contact us.

Summary of major concerns:

All three reviewers appreciate the application of unnatural amino acids to study *Salmonella* secretion in vitro and felt that it will be of interested to groups studying secretion mediated virulence in bacterial systems. Reviewer 3, in particular, was enthusiastic about development of this approach for analysis of bacterial pathogenesis.

At the same time, all reviewers agreed that while application of an established approach for analysis of pathogenesis in situ is a technical advance, the study did not break new ground with regard to our understanding of *Salmonella* pathogenesis. Overall, the reviewers, as well as Dr. Storz and myself, felt that the lack of experiments utilizing unnatural amino acids to address important questions in the field, muted the study's potential impact and thus its appeal to the broad readership of *eLife*. Resubmission is an option should you elect to apply the technique to answer previously intractable questions related to *Salmonella* virulence.

*Reviewer #1:*

The use of affinity and fluorescent protein tags on secreted effectors is often challenging due to the interference of these tags with protein folding and translocation. In this manuscript by Singh et al. authors demonstrate that bioorthogonal labeling using genetic incorporation of unnatural amino acids (UAA) can be used to overcome these challenges. Authors focused on a SifA, a T3SS effector of *Salmonella*, and visualized it in the host cell either by direct site-specific labeling with fluorescence UAA referred to as Hco or by indirect labeling with azide-containing UAA, AzF, that can be further visualized using Click-fluorophores. Using this approach, the authors demonstrated co-localization of secreted SifA with *Salmonella*-induced filaments (SIF) as well as phenotypic heterogeneity of SifA expression within a host cell.

The idea of using UAA labeling to overcome limitations of fusion proteins is not new, but the fact that this labeling works in situ (when *Salmonella* is inside the host cell vacuole) is exciting and could benefit host-pathogen interactions field. In my opinion, however, some of the manuscript conclusions, as well as broader applicability of this approach, require additional controls related to the labeling specificity and phenotypic consequences of synthetic tRNA/aaRS expression.

1. While TAG is relatively rare, there are still hundreds of TAG-containing genes, and the tRNA/aaRS expression would result in off-target incorporation of UAA.

a) how many genes in *Salmonella* contain TAG stop codon? Are there any effector proteins/virulence-related genes among them?

b) Figure 2A. A full fluorescence imaged gel containing stained lysates of pEVOL strains with either SifA WT and E52TAG need to be provided to demonstrate the extent of off-target labeling

c) Relevant to all experiments: Because other proteins can be labeled with UAA, the proper control is not cultures without UAA, but a WT SifA strain with pEVOL plasmid supplied with UAA. Only in such case, one can establish that fluorescence signal is specific to SifA and not other proteins (see related comment 3)

2. The potential of broader applicability of this approach to study host-microbe interaction is one of the selling points of the manuscript. However, it is unclear whether the synthetic system for tRNA/aaRS amber suppression (pEVOL) has negative consequences on virulence. For example, it is known that tRNA/aaRS expression is toxic in *E. coli* and results in growth defect and as well as other phenotypes. The effect of pEVOL plasmids on *Salmonella* on bacterial survival, virulence, timing of events is not established. All experiment are performed by comparing sifA- strain with sifA(52TAG) pEVOL strain, but the WT and WT pEVOL strains are not characterized. If pEVOL alters bacterial growth and survival, it will significantly limit the use of this approach for host-microbe interactions.

3. The conclusion of phenotypic heterogeneity of SPI-2 is not well-supported. The observed heterogeneity in fluorescence maybe not be physiologically relevant. It may arise as a result of the off-target labeling (see comment 1) or the use of the artificial two-plasmid system (pEVOL and p-sifA). The plasmid copy number varies among cells. In addition, the expression of tRNA/aaRS construct is driven by the leaky expression of arabinose inducible pBAD promoter. This promoter is metabolically regulated, and the expression heterogeneity is well-established in the literature.

*Reviewer #2:*

Due to previous difficulty in localizing the *Salmonella* secreted protein SifA during an infection, the authors employed a new approach by incorporating fluorescent or non-fluorescent unnatural amino acids in SifA via genetic code expansion. Utilizing this strategy, they were able to visualize SifA and showed that it co-localized with LAMP1, SseJ and kinesin on SIFs. In addition, they demonstrated that the T3SS SPI-2 is expressed heterogeneously (only ~16%) within the SCV (*Salmonella* containing vacuole).

1. Key figures in the paper (e.g. Figures 2 and 5) only show one or a few cells and no statistics were provided.

2. Previous studies have shown that SifA interacts with kinesin (e.g. Boucrot et al. 205 Science). As a result, demonstrating that SifA co-localizes with kinesin during an infection is confirmatory and provides a limited advance in our understanding of how SIFs form or function.

3. The authors previously demonstrated that ~30% of the bacterial cells express SPI-2 after acid exposure in vitro (Chakraborty et al. 2015 Plos Biol). Therefore, confirming that only a subpopulation expresses SPI-2 during an infection is also of limited significance.

*Reviewer #3:*

This is an interesting manuscript that nicely applies the application of nonsense codon suppressors that insert unnatural amino acids with reactive or fluorescent properties that allow tagging of bacterial proteins following injection into host cells. Although developed many years ago, the approach of inserting unnatural amino acids into proteins produced by pathogens to monitor their location and interactions within host cells is a wonderful addition to the toolkit for visualizing virulence factors, and will stimulate many other applications.

The authors make a strong argument for problems with previously used visualization methods, including use of bulky protein-tags or immunological approaches. Overall the concept, methodology, data, and images provide compelling evidence for the localization of SifA and the interactions of SifA with SseJ, SKIP, and kinesin in vivo without the use of approaches that are prone to artifacts.

I have a few questions and comments that are probably minor, but addressing these points will make the manuscript clearer.

1. UAA is an abbreviation for a nonsense codon. As the approach used to insert unnatural amino acids is based upon suppression of another nonsense codon, UAG, it is confusing to use "UAA" as an abbreviation for "unnatural amino acids". Something like u-AA or an alternative be better.

2. The initial sentences for two of the paragraphs of the Results section are redundant with the Introduction. Deletion of these redundancies would improve the readability of the manuscript. Lines 102-104 "Currently.… (See Introduction)."; and lines 123-125 "Intracellular.… (14-16)."

3. Note that an important point of this manuscript expressed on line 254 is that the results are based upon "native levels of protein expression". However, the proteins are expressed from the plasmid psifA52TAG, and the copy number of the plasmids are not indicated either in the text or in the Methods. As an important point of the manuscript, the reader should not have to go back to the original references to figure out the copy number of the plasmids or how the protein is expressed from the plasmid.

4. Isn't a caveat of the conclusion that this process is "bistable" dependent upon all of the *Salmonella* properly expressing SifA from the plasmid? Do you know that none of the *Salmonella* lost the plasmid during the subculture step described in line 322?

[Editors’ note: further revisions were suggested prior to acceptance, as described below.]

Thank you for submitting your article "Genetic code expansion enables visualization of *Salmonella* type three secretion system components and secreted effectors" for consideration by *eLife*. Your article has been reviewed by 2 peer reviewers, and the evaluation has been overseen by a Reviewing Editor and Gisela Storz as the Senior Editor. The following individual involved in review of your submission has agreed to reveal their identity: Kelly Hughes (Reviewer #2).

Essential revisions:

In addition to the revisions for clarity suggested by reviewer 2, please pay special attention to the request for the addition of statistical analysis to figures containing micrographs and a higher resolution image for Figure S13 as requested by reviewer 1. The addition of statistical analysis was a primary request of previous reviewers and was not addressed in the revised manuscript. While the current reviewers appreciate that the selected images represent hundreds of data points, statistics are necessary to provide insight into specifically how well "representative images" reflect the entire data landscape. Statistical analysis of micrograph data is standard practice and should not be too onerous. Please see note regarding the explanation of statistical analysis at the end of this email under "Information to help as you revise your submission" for more details.

*Reviewer #1:*

The authors have done a good job of responding to the majority of the reviewers' comments.

1. I do not think the authors adequately responded to the request for statistical data made by previous reviewers.

2. The data in Figure S13 showing that SsaP was mostly localized to the poles during early infection times is low resolution and not particularly convincing. Please replace with a higher resolution image and include statistical data on the fraction of cells with polar localization.

*Reviewer #2:*

The manuscript investigates the characterization by super-resolution fluorescence microscopy *Salmonella*-induced filament formation following engulfment by host cells and the role of SsaP in the *Salmonella* pathogenicity island 2 (SPI2) secretion-specificity switch from injectisome structure and assembly components to translocon and effector protein secretion. This work focuses on the SifA and SseJ effectors, which catalyzes the formation of *Salmonella* induced filaments. This is significant because in all type III secretion (T3S) systems characterized, the SsaP homologs act as molecular rulers to measure either injectisome length in the injectisome T3S systems or hook length in the flagellar T3S systems. The size of SsaP strongly argues against it acting as a ruler. The SsaP homologs include an ~300 N-terminal domain, which includes the T3S signal for export and the conserved, ~100 amino acid T3S4 domain described by Agrain et al. 2005, which catalyzes the T3S-specificity switch. SsaP at 124 amino acids in length contains the conserved T3S4 domain and whether or not SsaP secretion is required for the T3S-specificity switch is not known. Regardless, SsaP is unique among the T3S-specificity switch homologs making it a subject of great interest with the potential for unique mechanisms of action.

Overall, this represents a real tour-de-force and a significant contribution in understanding the SPI-2 mechanism of SIF formation and the role of SsaP in the secretion-specificity switch. The authors addressed the three original reviewers' concerns fully.

The Results section "Engineering and selection of label positions" should be retitled. It's a logical discussion for the methods used in this study. The authors make very good arguments as to why, under SPI2 inducing conditions, ncAA labeling of SifA and SsaP should work, and later they show it does. I just can't think of the title I would rather see (not helpful).

The second chapter of the Results section covers the GCE labeling. Unfortunately, the SsaP-mCherry fusion was cleaved. It's not unusual for this to occur with fusion proteins, but it does make life more difficult. The TAG at position 52 of SsaP, logically chosen, was successfully labeled as was SifA. Figure 2 clearly shows successful labeling of both proteins.

The third chapter demonstrates only a slight detrimental effect of the pEVOL – this must be an amber suppression plasmid, but it's not defined – on *Salmonella* pathogenesis. In general, when one compares a WT to a plasmid expression system the better control is the plasmid vector only control.

The fourth chapter demonstrates the complementation of the sifA deletion by engineered SifA-E52AzF – nice. Figure 3 imaging is spectacular.

Chapter 5 describes the fluorescence imaging of SifA-E52AzF. With appropriate controls the authors could visualize SifA within the filaments. This demonstrates that AzF labeling can clearly detect and visualize secreted effector protein.

Chapter 6 describes co-localization studies. SifA co-localizes with LAMP1. SifA, SseJ and kinesin co-localize. The controls are well done.

The next chapter (7) demonstrates that unlike SseJ, which forms continuous filaments, SifA shows punctate labeling.

Chapter 8 confirms SifA labeling at E52 with Hco behaves similar to the same with Azf.

In Chapter 9 we see a comparison of GCE-based labeling to antibody labeling that shows that GCE-based labeling is superior.

In Chapter 10 we get to SsaP. As one would predict from loss of a secretion-specificity switch protein, loss of SsaP was defective in effector secretion. SseB translocon was not secreted in the absence of SsaP indicating the switch occurs after needle completion and before translocon assembly. C-terminal fusion of mCherry to SsaP resulted in a functional SsaP, whereas an N-terminal fusion did not. This is consistent with what is observed with similar fusions to other T3SS SsaP homologs – except the SsaP-mCherry fusion is not stable and the mCherry domain is cleaved. This was circumvented with HA-tag insertions that produced functional SsaP-HA tagged derivatives. To verify that HA-SsaP was secreted into the HeLa cell cytosol, SsaP was labeled at position 65 with TCO*Lys, shown to be functional. Figure 5 does suggest that SsaP is secreted. There are very few molecules of SsaP representing a significant achievement in visualization studies.

---

## [Author Response]

[Editors’ note: the authors resubmitted a revised version of the paper for consideration. What follows is the authors’ response to the first round of review.]

Summary of major concerns:All three reviewers appreciate the application of unnatural amino acids to study *Salmonella* secretion in vitro and felt that it will be of interested to groups studying secretion mediated virulence in bacterial systems. Reviewer 3, in particular, was enthusiastic about development of this approach for analysis of bacterial pathogenesis.At the same time, all reviewers agreed that while application of an established approach for analysis of pathogenesis in situ is a technical advance, the study did not break new ground with regard to our understanding of *Salmonella* pathogenesis. Overall, the reviewers, as well as Dr. Storz and myself, felt that the lack of experiments utilizing unnatural amino acids to address important questions in the field, muted the study's potential impact and thus its appeal to the broad readership of eLife. Resubmission is an option should you elect to apply the technique to answer previously intractable questions related to *Salmonella* virulence.Reviewer #1:[…] 1. While TAG is relatively rare, there are still hundreds of TAG-containing genes, and the tRNA/aaRS expression would result in off-target incorporation of UAA.a) how many genes in *Salmonella* contain TAG stop codon? Are there any effector proteins/virulence-related genes among them?

We thank the reviewer for raising this excellent point that we had neglected to include in the manuscript. *Salmonella* Typhimurium (LT2) contains ~452 endogenous TAG stop codons out of 4696 genes. This means that 9.6% of its genome uses the TAG termination codon. Among them, there is one gene in the SPI-1 regulon (SopA) and 3 genes in SPI-2 (PipB, PipB2 and SspH2) that use TAG. Given this nearly 10%, one would anticipate a certain level of UAA (ncAA) incorporation at endogenous TAGs, which would lead to spurious dye labeling of other proteins in addition to the labeling of the protein of interest. However, given we are examining SifA under SPI-2 inducing conditions, we would expect that SPI-1 effectors to be present at very low (if any) concentrations. What about the other ancestral genes?

There are several important points to make in addressing this point:

1. It turns out that the position of the amino acid in the primary sequence and the neighboring base pairs in the mRNA influence the efficiency of amber suppression and thus ncAA incorporation [*Chembiochem*. 2016, 17, 1250-1256; *Curr Opin Chem Biol.* 2018, 46,146-155]; therefore, both the design of the TAG mutation, as well as its position is important. We have included sentences to this effect in the revised manuscript (see lines 143-165 and Figure 2).

2. It turns out that off-target labelling is really not a problem. Others engineered *E. coli* so that no other TAG codons were present except for one that was engineered in *ompC*, and surprisingly, off target labelling was not noticeably different in the engineered strain compared to a WT, non-recoded strain [*ACS Synth Biol.* 2017, 6, 233]. The authors speculated that either endogenous TAG codons are inefficiently decoded by the ncAA-pEVOL (in our case AzF-pEVOL), or the products are degraded. We basically see the same thing (see point 3). We now add a paragraph that deals with this issue (lines 155-159).

3. We evaluated the specificity of ncAA incorporation and labelling by constructing a GFP-fusion to SifA so that we could compare GFP fluorescence with fluorescence labelling at the 52-TAG position by Alexa 555. You can see in the first row of figures that the WT SifA-GFP is clearly visible, but when the 52-TAG-GFP construct is examined, in the absence of ncAA, there is no GFP (dark panel, right), because the stop codon truncates and no GFP is produced. In the second row, we add ncAA (AzF + pEVOL) to the WT SifA and examine Alexa555 and there is no labelling, because there is no position for labelling in the WT SifA construct. If off target labelling were significant, it would be visible in this field, but there is no signal and essentially no background. Similarly, with the 52-TAG-GFP construct, there is no background when there is no AzF added. Finally, when we examine 52-TAG-GFP and add AzF + pEVOL, (third row), there is a substantial GFP signal (as the amber is suppressed) and 52-TAG is labelled by AzF, as apparent by red fluorescence. Note that there is also really strong agreement with the red and the green fluorescence in the two panels. Thus, off target labelling does not contribute to the signal that we observe. This figure and a discussion about the findings are now included in the revised manuscript (lines 189-198, Figure 2—figure supplement 4).

4. Lastly, TAG stop codons are predominantly found in genes expressed at low levels [JBC 2014, 289, 30334-30342], so if present, they would be less visible.

b) Figure 2A. A full fluorescence imaged gel containing stained lysates of pEVOL strains with either SifA WT and E52TAG need to be provided to demonstrate the extent of off-target labeling

As mentioned above, off-target labelling was not an issue. Attempts to directly identify genomic off target incorporation of ncAAs did not provide evidence for their existence. Studies show no off-target incorporation in bacteria [*ACS Synth. Biol.* 2017, 6, 233; *Biochemistry*, 2017, 56,2161- 2165], nor in mammalian cells [*J. Am. Chem. Soc.* 2015,137,4602-4605], while others show incorporation into a small subset of TAG terminating genes that are only detectable in RF1 knockout strains [*Nat. Chem. Biol.* 2011, 7, 779-786; *Science,* 2013, 342, 357-360, 35; *Sci. Rep.* 2015, 5. 9699]. In figure 2A, SDS gels were run directly from the click reaction mixture without further washing the excess dye. We observed a few very low-level background fluorescence in figure 2A possibly because of a non-specific attachment of free dye molecules to cell proteins.

c) Relevant to all experiments: Because other proteins can be labeled with UAA, the proper control is not cultures without UAA, but a WT SifA strain with pEVOL plasmid supplied with UAA. Only in such case, one can establish that fluorescence signal is specific to SifA and not other proteins (see related comment 3)

We thank the reviewer for pointing out the missing information. To establish that the fluorescence signal observed in the host cell is specific to secreted SifA only, we infected HeLa cells with wild-type *Salmonella* in the presence of pEVOL and ncAA. To visualize if any ncAA incorporated effectors secreted into host cells, HeLa cells were treated with BDP dye as described in the methods. After labeling, excess dye was washed out and cells were fixed with 4% PFA and immunostained for SseJ after 16 h post-infection. The fluorescence microscopy image shows the absence of intracellular fluorescent signals in the green channel, and non-specific fluorescence signals were nearly absent. We have included this results in the revised manuscript (see line 228-240; *SI Appendix* Figure S7).

2. The potential of broader applicability of this approach to study host-microbe interaction is one of the selling points of the manuscript.

We agree! And we think it is even stronger now.

However, it is unclear whether the synthetic system for tRNA/aaRS amber suppression (pEVOL) has negative consequences on virulence. For example, it is known that tRNA/aaRS expression is toxic in E. coli and results in growth defect and as well as other phenotypes. The effect of pEVOL plasmids on *Salmonella* on bacterial survival, virulence, timing of events is not established. All experiment are performed by comparing sifA- strain with sifA(52TAG) pEVOL strain, but the WT and WT pEVOL strains are not characterized. If pEVOL alters bacterial growth and survival, it will significantly limit the use of this approach for host-microbe interactions.

The reviewer raises an excellent point here. Although high suppression efficiency was achieved with the amber stop codon, no significant adverse effects on bacterial growth could be observed in *E. coli* with 7–8% of genes terminating in TAG [*Nat. Chem. Biol.,* 2017, 13, 446-450; *ACS Synth Biol.* 2017, 6, 233], nor did it change the growth rate in *Synechococcus elongatus* with 40% of genes terminating in TAG [Biochemistry, 2017, 56,2161- 2165]. Nonetheless, to examine the consequence of orthogonal amber suppression (pEVOL), we addressed this point in two ways, we first performed a simple growth curve comparing the WT strain +/- pEVOL (*SI Appendix* Figure S5A). Next, we examined the effect of pEVOL on virulence by infecting HeLa cells with WT +/-pEVOL and enumerating intracellular bacteria at 2 and 16h post-infection. The results indicated that there was a slight reduction in the intracellular replication (74% compared to no plasmid at 100%). Together, these results suggest that orthogonal tRNA/aaRS amber suppression (pEVOL) system has no negative consequences on the pathogenesis of *Salmonella*. (see lines 199-209 and *SI Appendix,* Figure S5A, B).

3. The conclusion of phenotypic heterogeneity of SPI-2 is not well-supported. The observed heterogeneity in fluorescence maybe not be physiologically relevant. It may arise as a result of the off-target labeling (see comment 1) or the use of the artificial two-plasmid system (pEVOL and p-sifA). The plasmid copy number varies among cells. In addition, the expression of tRNA/aaRS construct is driven by the leaky expression of arabinose inducible pBAD promoter. This promoter is metabolically regulated, and the expression heterogeneity is well-established in the literature.

We strongly disagree with the reviewer on this point of SPI-2 heterogeneity, however we do agree about the problems of using pBAD. Previous in vitro studies [Chakraborty et al. 2015 Plos. Biol.] from our lab have demonstrated the heterogeneity, but that is not the focus of this manuscript, so we have removed this section.

Reviewer #2:Due to previous difficulty in localizing the *Salmonella* secreted protein SifA during an infection, the authors employed a new approach by incorporating fluorescent or non-fluorescent unnatural amino acids in SifA via genetic code expansion. Utilizing this strategy, they were able to visualize SifA and showed that it co-localized with LAMP1, SseJ and kinesin on SIFs. In addition, they demonstrated that the T3SS SPI-2 is expressed heterogeneously (only ~16%) within the SCV (*Salmonella* containing vacuole).1. Key figures in the paper (e.g. Figures 2 and 5) only show one or a few cells and no statistics were provided.

Representative images are shown from many tens of infections we have performed. We now state: Infections were performed tens of times, hundreds of cells were imaged per infection and the figures display representative images in the Methods and reiterate this fact in the figure legends. Figure 5 is now Figure 4 in the revised manuscript and the number of cells used for analysis is in the legend.

2. Previous studies have shown that SifA interacts with kinesin (e.g. Boucrot et al. 205 Science). As a result, demonstrating that SifA co-localizes with kinesin during an infection is confirmatory and provides a limited advance in our understanding of how SIFs form or function.

It is disappointing to us that the reviewer failed to see the importance of the labelling and the application of the method. In my conversations with colleagues, many of them have tried (and failed!) to use GCE to site-specifically label their virulence factors in a number of pathogens. We succeeded in labeling SifA, which was intractable to labeling by other approaches. The focus of the manuscript was not on revealing new aspects of the Sifs, although we did discover that while SseJ is continuously labelled along the Sifs and subject to fixation artifacts [Gao et al., mBio 2018], SifA labelling is punctate, independent of the fixation method. This result means that SifA is not directly force-dependent and the stoichiometry is not 1:1 SifA to SseJ. However, the real point was to show that we could label a protein using GCE that we had been unable to label by other means. The revised manuscript further emphasizes the utility of the approach by labelling another protein (SsaP) that was also impossible to label by constructing fusion proteins, since the fusion protein was cleaved (see *SI Appendix,* Figure S1). We provide new information that demonstrates the importance of SsaP in virulence and show for the first time that it is secreted during infection (*SI Appendix,* Figure S1, S2, S10-11 and revised manuscript Figure 5).

3. The authors previously demonstrated that ~30% of the bacterial cells express SPI-2 after acid exposure in vitro (Chakraborty et al. 2015 Plos Biol). Therefore, confirming that only a subpopulation expresses SPI-2 during an infection is also of limited significance.

We respectfully disagree with the reviewer on this point, Chakraborty et al. 2015 only showed in vitro heterogeneity and we think it is important to establish the presence of cheaters in the vacuole. However, we eliminated this section based on the criticism

from the other reviewers (point #3, above).

Reviewer #3:This is an interesting manuscript that nicely applies the application of nonsense codon suppressors that insert unnatural amino acids with reactive or fluorescent properties that allow tagging of bacterial proteins following injection into host cells. Although developed many years ago, the approach of inserting unnatural amino acids into proteins produced by pathogens to monitor their location and interactions within host cells is a wonderful addition to the toolkit for visualizing virulence factors, and will stimulate many other applications.The authors make a strong argument for problems with previously used visualization methods, including use of bulky protein-tags or immunological approaches. Overall the concept, methodology, data, and images provide compelling evidence for the localization of SifA and the interactions of SifA with SseJ, SKIP, and kinesin in vivo without the use of approaches that are prone to artifacts.I have a few questions and comments that are probably minor, but addressing these points will make the manuscript clearer.1. UAA is an abbreviation for a nonsense codon. As the approach used to insert unnatural amino acids is based upon suppression of another nonsense codon, UAG, it is confusing to use "UAA" as an abbreviation for "unnatural amino acids". Something like u-AA or an alternative be better.

Ok. We have modified UAA to ncAA throughout the manuscript.

2. The initial sentences for two of the paragraphs of the Results section are redundant with the Introduction. Deletion of these redundancies would improve the readability of the manuscript. Lines 102-104 "Currently.… (See Introduction)."; and lines 123-125 "Intracellular.… (14-16)."

Done.

3. Note that an important point of this manuscript expressed on line 254 is that the results are based upon "native levels of protein expression". However, the proteins are expressed from the plasmid psifA52TAG, and the copy number of the plasmids are not indicated either in the text or in the Methods. As an important point of the manuscript, the reader should not have to go back to the original references to figure out the copy number of the plasmids or how the protein is expressed from the plasmid.

We now include this information in the Methods and note that *sifA* is expressed behind its native promoter.

4. Isn't a caveat of the conclusion that this process is "bistable" dependent upon all of the *Salmonella* properly expressing SifA from the plasmid? Do you know that none of the *Salmonella* lost the plasmid during the subculture step described in line 322?

This section has been removed as requested by Reviewers 1 and 2.

[Editors’ note: what follows is the authors’ response to the second round of review.]

Reviewer #1:The authors have done a good job of responding to the majority of the reviewers' comments.1. I do not think the authors adequately responded to the request for statistical data made by previous reviewers.

According to the suggestions and comments of current and previous reviewer, we now included extensive statistical analysis (see Figure 2—figure supplement 5, Figure 3—figure supplement 4, Figure 3—figure supplement 5 and Figure 4—figure supplement 1).

2. The data in Figure S13 showing that SsaP was mostly localized to the poles during early infection times is low resolution and not particularly convincing. Please replace with a higher resolution image and include statistical data on the fraction of cells with polar localization.

We have updated Figure S13 with high resolution image along with statistical data on the fraction of cells containing polarly localized SsaP (now Figure 2D and Figure 2—figure supplement 5).